# SwiftKV: Fast Prefill-Optimized Inference with Knowledge-Preserving Model Transformation

## Abstract

LLM inference for popular enterprise use cases, such as summarization, RAG, and code-generation, typically observes orders of magnitude longer prompt lengths than generation lengths. This characteristic leads to high cost of prefill and increased response latency. In this paper, we present SwiftKV, a novel model transformation and distillation procedure specifically designed to reduce the time and cost of processing prompt tokens while preserving high quality of generated tokens. SwiftKV combines three key mechanisms: i) SingleInputKV, which prefills later layers' KV cache using a much earlier layer's output, allowing prompt tokens to skip much of the model computation, ii) AcrossKV, which merges the KV caches of neighboring layers to reduce the memory footprint and support larger batch size for higher throughput, and iii) a knowledge-preserving distillation procedure that can adapt existing LLMs for SwiftKV with minimal accuracy impact and low compute and data requirement. For Llama-3.1-8B and 70B, SwiftKV reduces the compute requirement of prefill by 50% and the memory requirement of the KV cache by 62.5% while incurring minimum quality degradation across a wide range of tasks. In the end-to-end inference serving using an optimized vLLM implementation, SwiftKV realizes up to $2\times$ higher aggregate throughput and 60% lower time per output token. It can achieve a staggering 560 TFlops/GPU of normalized inference throughput, which translates to 16K tokens/s for Llama-3.1-70B in 16-bit precision on $4\times$ H100 GPUs. Our training, inference, and model implementations are open-sourced at https://anonymized.link.

## 1 Introduction

Large Language Models (LLMs) are quickly becoming an integral enabler of enterprise applications and offerings, including code and data co-pilots (Chen et al., 2021; Pourreza & Rafiei, 2024), retrieval augmented generation (RAG) (Lewis et al., 2020; Lin et al., 2024), summarization (Pu et al., 2023; Zhang et al., 2024), and agentic workflows (Wang et al., 2024; Schick et al., 2023). While it is clear that LLMs can add value to these applications, the cost and speed of inference determine their practicality. Therefore, improving the aggregate throughput and reducing latency of LLM inference has become an increasingly important topic of interest, with various efforts (Sec. 2) tackling the problem from multiple angles.

In this paper, we take a unique approach to improving LLM inference for enterprise applications based on the key observation that typical enterprise workloads process many more input tokens than output tokens. For example, tasks like code completion, text-to-SQL, summarization and RAG each submit long prompts but produce a small number of generated tokens, and a majority of enterprise LLM use cases incur a 10:1 ratio between prompt and generated tokens.[1]

Based on this observation, we designed *SwiftKV*, which improves throughput and latency by: i) reducing the computation required to pre-fill the KV cache for input tokens, and ii) enabling memory savings to support larger batch sizes needed to serve LLMs more cost effectively (Sheng et al., 2023; Pope et al., 2022; Yu et al., 2022). SwiftKV (Fig. 1) consists of four key components:

**SingleInputKV.** SingleInputKV rewires an existing model so that the pre-fill stage during inference can skip a number of later layers in the network, and their KV cache are computed by a single earlier layer. SingleInputKV is motivated by the observation that the output hidden states of the later transformer layers do not change significantly (see Sec. 3.2, also independently discovered by Liu et al. (2024c)). With

---

[1]Exact use cases hidden to preserve anonymity but will be included in final paper.

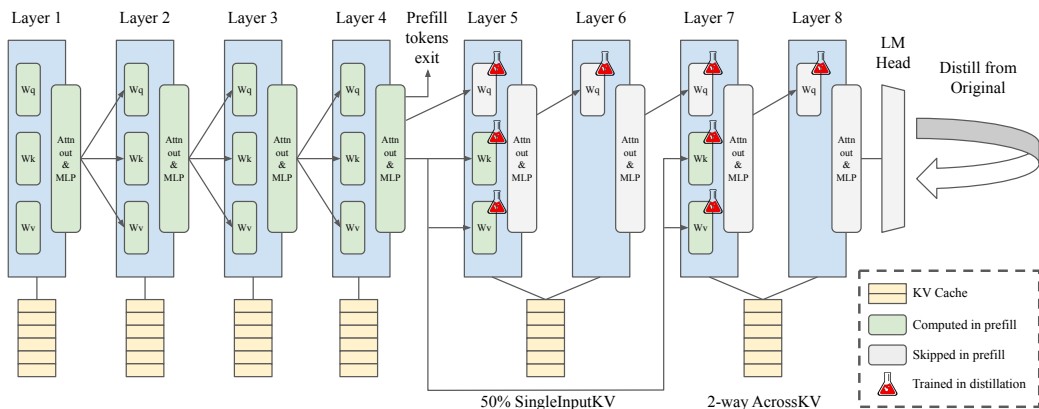

Figure 1: Illustration of SwiftKV with 50% SingleInputKV and 2-way AcrossKV. After distillation, the KV cache of layers 5–8 can all be populated using the hidden state outputs of layer 4. For prefill tokens, the query, attention, and MLP operations of layers 5–8 may be skipped altogether, while decode tokens complete all layers. Existing models may be efficiently adapted for SwiftKV by distilling from the original unmodified model using a small dataset. Model knowledge is preserved by keeping the trainable parameters limited to the Q, K, and V projections of the layers affected by SingleInputKV.

SingleInputKV, the computation required for pre-fill is reduced by approximately the number of layers skipped. We found that it is possible to skip at least 50% of the layers without significantly impacting the model quality (Sec. 4.2), which translates to a 50% reduction of the pre-fill computation in inference.

**AcrossKV.** While SingleInputKV reduces the pre-fill computation, it does not reduce the KV cache memory requirement. AcrossKV combines the KV projections from multiple adjacent layers into a single one, and share the KV cache across these layers to reduce its size in memory. AcrossKV allows significant memory savings, which unlocks higher throughput by enabling larger batches during inference. In Sec. 3, we show that AcrossKV can reduce the KV cache size by 25% with less than a 1% quality gap. We also show in ablation studies (Sec. 5) that AcrossKV is compatible with existing KV cache quantization methods, which combine to unlock 62.5% reduction in KV cache size.

**Knowledge Recovery.** Although SingleInputKV and AcrossKV can be applied to existing LLMs with minimal changes to their architectures, we found that the resulting model parameters should still be adapted to the new architecture to recover their original prediction quality. This can be done via distillation from the original model. With SwiftKV, we found that a lightweight distillation is sufficient, with only a fraction of the model (Q, K, and V projections of the affected layers) trained on 680M tokens, which takes less than 3 hours on a single $8\times$ H100 node for Llama-3.1-8B.

**SwiftKV Optimized Inference.** To realize the computation and memory reductions of SingleInputKV and AcrossKV into end-to-end throughput and latency improvements, we implemented SwiftKV in vLLM (Kwon et al., 2023). Our implementation includes several additional optimizations, including fusing all KV-projections beyond layer $l$ into a single GEMM operation, and integrated memory management needed to lower the KV cache memory footprint achievable via AcrossKV.

SwiftKV increases the aggregate throughput of enterprise workloads by up to $2\times$, while reducing time-to-first-token (TTFT) and time-per-output-token (TPOT) by up to 50% and 60%, respectively. In fact, for Llama-3.1-70B, SwiftKV can achieve a normalized throughput of 560 TFLops/GPU[2]. This is an unprecedented 56.6% MFU utilization for inference (Sec. 4.3). We show that SwiftKV incurs minimal quality degradation averaged across a wide range of tasks (Sec. 4.2), including ARC-Challenge (Clark et al., 2018), Winogrande (Sakaguchi et al., 2019), HellaSwag (Zellers et al., 2019), TruthfulQA (Lin et al., 2022), MMLU (Hendrycks et al., 2021), and GSM8K (Cobbe et al., 2021).

In addition to these main results, in Sec. 5 we discuss the impact of distillation, datasets, choice of trainable parameters for training SwiftKV. We also present our analysis of the hidden state similarities, and how AcrossKV can be extended and combined with other KV cache compression

---

[2]Normalized throughput and MFU is based on number of floating point operations in the baseline model.

works. Additionally, we also discuss how SingleInputKV can enable compute savings not just during pre-fill but also during decoding phase.

Lastly, we open-sourced the training and inference code for SwiftKV that can be used to fully reproduce our results at `https://anonymized.link`, as well as several SwiftKV models that can be used directly by the community at `https://anonymized.link`.

## 2 RELATED WORK

**Hardware and System Optimizations.** Lower-precision quantization like FP8 (Kuzmin et al., 2024) can enable the use of tensor-cores to accelerate inference (Luo et al., 2024). System approaches like PagedAttention (Kwon et al., 2023), Tensor-Parallelism(Shoeybi et al., 2020), Split-Fuse (Holmes et al., 2024; Agrawal et al., 2024), FlashAttention (Dao et al., 2024), and their optimized implementations in TensorRT (NVIDIA, 2019), FasterTransformer (NVIDIA, 2021), vLLM (Kwon et al., 2023), and DeepSpeed-Inference (Aminabadi et al., 2022) enable better parallelization, batching, and scheduling to eliminate performance overheads and achieve better hardware peak utilization without impacting model quality. In contrast, SwiftKV is a model architecture optimization and is complementary to these works.

**Sparse attention optimizations.** Systems such as ALISA (Zhao et al., 2024) and MInference (Jiang et al., 2024) leverage naturally-occurring sparsity patterns in transformer models to reduce the computation of the quadratic attention operation. Particularly, like SwiftKV, MInference also targets the prefill phase of inference. Sparse attention can be particularly effective for very long sequence lengths (e.g. 100K - 1M tokens) when attention is the dominant operation. In comparison, SwiftKV reduces prefill computation by skipping not just the attention operation, but also the query/output projections and MLP of certain layers. This means that (1) SwiftKV can be more suited for inputs with more moderate lengths when MLP is the dominant operation, and (2) SwiftKV, which either runs attention as-is or skips attention entirely, is complementary to sparse attention methods which are more concerned with the implementation of attention itself.

**Memory Compression.** A wide range of techniques have been developed to reduce the memory need of inference. Lower-precision quantization techniques like FP8/FP4 can reduce the memory for both KV cache and parameters (Hooper et al., 2024). Attention optimization techniques like MQA (Shazeer, 2019), GQA (Ainslie et al., 2023b), low-rank attention (Chang et al., 2024) also reduce the KV Cache. These approaches are complementary to SwiftKV, which we demonstrate in Sec. 4.2 and Sec. 5.3. Like AcrossKV, MiniCache (Liu et al., 2024b) also considers merging the KV cache of consecutive layers. However, AcrossKV enables consolidating more than just two layers, allowing for higher level of compression, and does not require any token retention strategy where distinct KV caches are stored for special tokens, allowing for simpler implementation.

Furthermore, while many of these approaches only focus on reducing the memory, SwiftKV can reduce both the prefill computation (via SingleInputKV) along with memory. As we show in Sec. 5.1, compute reduction rather than memory reduction is crucial for accelerating inference in compute-bound scenarios with sufficient memory, which is common in production with modern GPUs (H100, A100).

## 3 SWIFTKV: DESIGN AND IMPLEMENTATION

### 3.1 PRELIMINARIES

In transformer models (Vaswani et al., 2017), attention enables each token to focus on other tokens by comparing *queries* ($Q$) with *keys* ($K$) and using *values* ($V$) to compute the final representation. For a sequence of input tokens $x^{(1)},...,x^{(n)}$, the projections are defined as follows: $Q = XW_Q, \quad K = XW_K, \quad V = XW_V$, where $X \in \mathbb{R}^{n \times d}$ are the input embeddings, and $W_Q \in \mathbb{R}^{d \times d_k}$ and $W_K, W_V \in \mathbb{R}^{d \times d_g}$ are trained model parameters with $d_g | d_k$. Hereafter, we may also refer to $W_K$ and $W_V$ as a single matrix $W_{KV} \in \mathbb{R}^{d \times 2d_k}$.

During the *prefill phase* of inference, the model processes the entire input sequence at once, computing $K$ and $V$ for all tokens in parallel (or in chunks in the case of Split-Fuse (Holmes et al., 2024; Agrawal et al., 2024)). This typically occurs when the model handles an initial prompt or context.

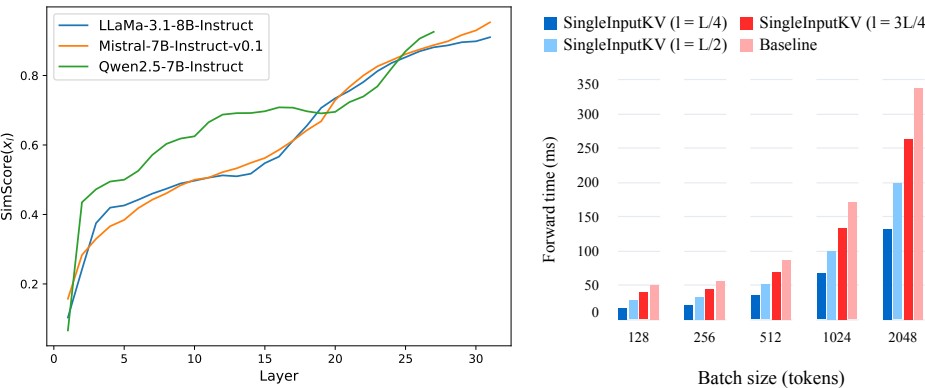

Figure 2: LEFT: the input similarity of several smaller scale models (Fig. A.1 in the Appendix shows a similar observation for larger models). RIGHT: The time per forward pass of Llama-3.1-8B. SingleInputKV effectively reduces the forward pass processing time across a range of batch sizes.

During the *decoding phase* of inference, new tokens are generated one at a time. When predicting the next token, only the query ($Q^{(t+1)}$) for the new token needs to be computed, while the model must attend to the keys and values ($K^{(1)},...,K^{(t)}, V^{(1)},...,V^{(t)}$) of all previously processed tokens.

To optimize efficiency in the decoding phase, *KV caching* is employed. After processing each token $t$, the newly computed $K^{(t)}$ and $V^{(t)}$ are stored in a cache. For the next token $t+1$, only the new query $Q^{(t+1)}$, key $K^{(t+1)}$, and value $V^{(t+1)}$ are computed. The attention computation will then utilize the cached $K$ and $V$ from all prior tokens, allowing for reduced computational overhead during inference.

### 3.2 SINGLEINPUTKV: PROJECT KV CACHE FROM A SINGLE LAYER

Assume the input of $l$-th layer is $\mathbf{x}_l$, and its $i$-th token is $\mathbf{x}_l^{(i)}$. Prior studies (Liu et al., 2024c; Gromov et al., 2024) showed that $\mathbf{x}_l$ becomes more similar as the depth grows. Here, we conduct a similar study.

We compute the average input similarity between $l$-th layer's input and all remaining layers' input, i.e.,

$$\text{SimScore}(\mathbf{x}_l) = \frac{\sum_{j=l+1}^{L}\text{Similarity}(\mathbf{x}_l,\mathbf{x}_j)}{L-l}, \tag{1}$$

where $L$ is the number of layers in the model and $\text{Similarity}(\mathbf{x}_l,\mathbf{x}_j)$ is the average cosine similarity between all $\mathbf{x}_l^{(i)}$ and $\mathbf{x}_j^{(i)}$ tokens.

We use 50 random training examples from `HuggingFaceH4/ultrachat_200k` to estimate $\text{SimScore}(\mathbf{x}_l)$, and the results of Llama-3.1-8B-Instruct, Mistral-7B-Instruct-v0.1, and Qwen2.5-7B-Instruct are shown in the left of Fig. 2. As the layers get deeper, $\text{SimScore}(\mathbf{x}_l)$ gets higher. Particularly, around half of the depth, the average similarity of $\mathbf{x}_l$ with $\mathbf{x}_{>l}$ is above 0.5 for all models, which shows that the difference of input hidden states are small in deeper layers.

Based on this observation, the first key component of SwiftKV is to use $l$-th layer's output $\mathbf{x}_{l+1}$ to compute the KV cache for all remaining layers. More specifically, SwiftKV retains the standard transformer architecture up to and including the $l$-th layer, but the KV cache for all remaining layers are computed immediately using $\mathbf{x}_{l+1}$, i.e.

$$\mathbf{KV}_j = \mathbf{W}_{KV}^j \mathbf{x}_{l+1}, \quad \text{for all } j > l, \tag{2}$$

where $\mathbf{KV}_j$ is the KV cache for $j$-th layer and $\mathbf{W}_{KV}^j$ is its KV projection weight matrix.

**Prefill Compute Reduction.** SingleInputKV can enable significant reduction in prefill compute during LLM inference. Originally, all input tokens must be processed by all transformer layers. With SingleInputKV, input tokens[3] only need to compute $\mathbf{W}_{KV}^j \mathbf{x}_{l+1}$ for layers $j > l$ to generate layer $j$'s

---

[3]The very last input token still needs to compute all layers to generate the first output token.

KV cache, and all other operations (i.e., QO projections, Attention, and MLP) of layers $j > l$ can be skipped entirely. When prefill computation dominates generated token computation, this reduces the total inference computation to approximately $l/L$. Fig. 1 illustrates the operations skipped by SingleInputKV, and Table 1 shows a more detailed example compute breakdown for Llama-3.1-70B.

### 3.3 ACROSSKV: SHARING KV CACHE FOR CONSECUTIVE LAYERS

GQA (Ainslie et al., 2023a), one of the most widely adopted KV cache compression methods, showed that the KV cache can be easily shared within a transformer layer. Later, Liu et al. (2024a) showed that the KV cache can be merged for certain pairs of adjacent layers. AcrossKV extends the ideas to cross-layer KV cache sharing.

Table 1: Breakdown of transformer operations for Llama-3.1-70B with SwiftKV (in GFlops per prefill token).

| Model | Vocab | K,V | Q,O | MLP | Attn. | Total | Rel. |
|---|---|---|---|---|---|---|---|
| Baseline | 4.3 | 2.6 | 22 | 113 | 160 | 302 | 100% |
| 25% SingleInputKV | 4.3 | 2.6 | 16 | 85 | 120 | 228 | 75.5% |
| 50% SingleInputKV | 4.3 | 2.6 | 11 | 56 | 80 | 154 | 51.0% |
| 50% SingleInputKV $+ 4\times$ AcrossKV | 4.3 | 1.7 | 11 | 56 | 80 | 153 | 50.7% |

Particularly, instead of computing KV cache for all of the remaining layers as shown in equation 2, AcrossKV selectively chooses one layer to compute the KV cache for several consecutive layers and share it within the small group. The key idea is shown in Fig. 1. As AcrossKV can combine multiple layers' KV caches into a single one rather than just two adjacent layers, it offers higher potential reduction ratio compared to Liu et al. (2024a) while simplifying the implementation to realize the benefits of the KV cache reduction. (See Sec. 2 for more detailed comparison with Liu et al. (2024a)).

### 3.4 KNOWLEDGE RECOVERY

While SingleInputKV preserves all the original parameters, it re-wires the architecture so that the KV cache projections may receive different inputs. We found that this re-wiring (and AcrossKV) requires fine-tuning to recover the original capabilities from the modified model. As we only change the computation of the attention part for layer $> l$, this can be achieved by fine-tune just the $\mathbf{W}_{QKV}$ weight matrices from the $(l+1)$-th layer onwards. However, instead of directly fine-tuning these parameters using standard LM loss, we find that distilling using the output logits of the original model allows for better knowledge recovery (see Sec. 5 for more details).

**Implementing the Distillation.** Since only a few $\mathbf{W}_{QKV}$ parameters need fine-tuning, we are able to do a memory efficient parameter-sharing based distillation. More specifically, we keep a single copy of the original model weights in memory that are frozen during training, and add an extra trainable copy of the $\mathbf{W}_{QKV}$ parameters for layers $> l$ initialized using the original model (See Fig. 1).

During the training, we create two forward modes for the later layers $> l$, one with original frozen parameters using original architecture, and another with the SwiftKV re-wiring using new QKV projections i.e.,

$$\mathbf{y}_{teacher} = \mathbf{M}(\mathbf{x}, SwiftKV = False), \text{ and } \mathbf{y}_{student} = \mathbf{M}(\mathbf{x}, SwiftKV = True), \qquad (3)$$

where $\mathbf{y}_{\cdot}$ is the final logits, $\mathbf{M}$ is the model, and $\mathbf{x}$ is the input. Afterwards, we apply the standard distillation loss ($L$) upon the outputs with temperature ($\tau$) using (Hinton et al., 2015). After the distillation, the original KV projection layers $> l$ are discarded during inference.

This method allows us to perform the distillation for Llama-3.1-8B-Instruct on 680M tokens of data in 3 hours using 8 H100 GPUs, and Llama-3.1-70B-Instruct in 5 hours using 32 H100 GPUs across 4 nodes.

### 3.5 OPTIMIZED IMPLEMENTATION FOR INFERENCE

LLM serving systems can be complex and incorporate many simultaneous optimizations at multiple layers of the stack, such as PagedAttention (Kwon et al., 2023), Speculative Decoding (Leviathan et al., 2023), SplitFuse (Holmes et al., 2024; Agrawal et al., 2024), and more. One benefit of SwiftKV is that it makes minimal changes to the model architecture, limited to only a few linear projection layers. This means that SwiftKV can easily be integrated into existing serving systems without implementing new kernels (e.g. for custom attention operations or sparse computation) or novel inference procedures.

**Implementation in vLLM.** To realize the performance benefits of SwiftKV, we integrated it with vLLM . Our implementation is compatible with vLLM's chunked prefill, which processes prefill tokens in chunks and may mix prefills and decodes in each minibatch. During each forward pass, after completing layer $l$, the KV-cache for the remaining layers ($>l$) are immediately computed, and only the decode tokens are propagated through the rest of the model layers.

**GEMM and Memory Optimizations.** Upon this basic implementation, we implemented two additional optimizations. First, *SingleInputKV fusion*: instead of computing the KV cache $\mathbf{KV}_j$ for each layer $j > l$ one at a time, we fused all $\mathbf{W}_{KV}^j$ into one large weight matrix $\mathbf{W}_{KV}^{j>l}$ so that their KV cache can be computed with a single efficient GEMM operation. Second, *AcrossKV reduction*: we modified vLLM to only allocate one layer's KV-cache for each group of merged layers, which realizes the memory gains of AcrossKV.

# 4 MAIN RESULTS

## 4.1 SETUP

**Training and Evaluation.** We use Llama-3.1-8B-Instruct and Llama-3.1-70B-Instruct as our base models for SwiftKV. Our training datasets include a mixture of the full supervised training data from `HuggingFaceH4/ultrachat_200k` (Ding et al., 2023) and `teknium/OpenHermes-2.5` (Teknium, 2023). We evaluated model quality using a modified LM-Eval-Harness (Gao et al., 2024)[4] due to its support for the custom prompt format of Llama-3.1, particularly for MMLU and MMLU-CoT (Hendrycks et al., 2021), GSM8K (Cobbe et al., 2021), and Arc-Challenge (Clark et al., 2018). For more details, please see Appendix B.

**Compression Metrics.** For prefill computation, we report the approximate reduction as $(L-l)/L$ due to SingleInputKV, and for KV cache, we report the exact memory reduction due to AcrossKV. For example, SwiftKV with SingleInputKV ($l = L/2$) and 4-way AcrossKV is reported as 50% prefill compute reduction and 37.5% KV cache memory reduction. We further study how these theoretical compute and memory reductions translate into end-to-end inference improvements in Sec. 4.3.

**Inference Performance.** In our inference evaluation, we focus on two common scenarios: *batch-inference* for cost sensitive scenarios and *interactive-inference* for latency sensitive scenario.

*Batch-Inference* When processing text in bulk or when serving a model under high usage demand, it is important to achieve high *combined throughput* in terms of input + output tokens processed per second. For bulk processing, the combined throughput determines the time it takes to finish the job. For interactive use, it determines the volume of concurrent users and requests that can be supported per unit of hardware. In both scenarios, the combined throughput is a key determinant of the cost of serving the model.

*Interactive-Inference* In interactive scenarios (e.g., chatbots, copilots), not only the combined throughput is important, but also metrics that define the end-user experience. Chief upon them are the time-to-first-token (TTFT) and time-per-output-token (TPOT). TTFT is the time between the user sending a message and receiving the first token in the response. TPOT is the time between each output token after the first token has been received. Low TTFT and TPOT are desirable by interactive applications to deliver smooth usage experiences.

For all experiments on Llama-3.1-8B-Instruct, we use 1 NVIDIA H100 GPU with 80GB of memory, and for all experiments on Llama-3.1-70B-Instruct, we use 4 NVIDIA H100 GPUs running the model with 4-way tensor parallelism. We provide the full hardware and vLLM configurations in Appendix B.2

## 4.2 MODEL QUALITY WITH PREFILL COMPUTE REDUCTION

**Llama-3.1-8B-Instruct.** The top rows of Table 2 show that SwiftKV can preserve model quality well until 50% prefill reduction using SingleInputKV. For 25% prefill reduction, the accuracy degradation is only about 0.12 points and for 50% reduction, the gap is about 1 point [5]. When we push to 62.5%

---

[4] `https://github.com/neuralmagic/lm-evaluation-harness/tree/llama_3.1_instruct`

[5] Note that we did not try to find the best training recipe, regarding to either training data (e.g., we did not include any math or coding datasets) or training pipeline (e.g., we did not include reinforce-learning rated steps,

Table 2: Llama-3.1-8B-Instruct

| Model | SingleInputKV (Prefill Reduction) | AcrossKV (Cache Reduction) | Arc-Challenge 0-shot | Winogrande 5-shots | HelloSwag 10-shots | truthfulqa 0-shot | MMLU 5-shots | MMLU-CoT 0-shot | GSM-8K 8-shots | Avg. |
|---|---|---|---|---|---|---|---|---|---|---|
| Baseline | N/A | N/A | 82.00 | 77.90 | 80.40 | 54.56 | 67.90 | 70.63 | 82.56 | 73.71 |
| SwiftKV | ✓(25%) | ✗ | 82.08 | 77.98 | 80.63 | 54.59 | 67.95 | 70.45 | 81.43 | 73.59 |
| SwiftKV | ✓(50%) | ✗ | 80.38 | 78.22 | 79.30 | 54.54 | 67.30 | 69.73 | 79.45 | 72.70 |
| SwiftKV | ✓(62.5%) | ✗ | 71.76 | 75.77 | 78.21 | 52.73 | 61.55 | 53.68 | 68.92 | 66.09 |
| SwiftKV | ✓(50%) | 2-way (25%) | 80.29 | 77.82 | 79.03 | 54.66 | 66.96 | 68.39 | 75.59 | 71.82 |
| SwiftKV | ✓(50%) | 4-way (37.5%) | 79.35 | 77.51 | 78.44 | 54.96 | 65.71 | 67.75 | 76.72 | 71.49 |
| SwiftKV | ✓(50%) | 8-way (43.75%) | 79.18 | 77.19 | 77.38 | 54.79 | 65.73 | 66.88 | 72.33 | 70.50 |
| SwiftKV | ✓(50%) | 16-way (46.875%) | 78.24 | 76.80 | 76.87 | 56.86 | 64.65 | 65.86 | 72.25 | 70.22 |

Table 3: Llama-3.1-70B-Instruct

| Model | SingleInputKV Prefill Reduction | AcrossKV Cache Reduction | Arc-Challenge 0-shot | Winogrande 5-shots | HelloSwag 10-shots | truthfulqa 0-shot | MMLU 5-shots | MMLU-CoT 0-shot | GSM-8K 8-shots | Avg. |
|---|---|---|---|---|---|---|---|---|---|---|
| Baseline | N/A | N/A | 93.34 | 85.16 | 86.42 | 59.95 | 83.97 | 86.21 | 95.15 | 84.31 |
| SwiftKV | ✓(25%) | ✗ | 93.00 | 84.69 | 85.98 | 59.43 | 82.82 | 85.81 | 95.07 | 83.83 |
| SwiftKV | ✓(50%) | ✗ | 93.09 | 83.82 | 84.45 | 58.40 | 82.51 | 85.00 | 93.56 | 82.98 |
| SwiftKV | ✓(50%) | 2-way (25%) | 92.92 | 82.95 | 84.10 | 57.79 | 82.66 | 84.55 | 93.48 | 82.63 |
| SwiftKV | ✓(50%) | 4-way (37.5%) | 92.92 | 83.74 | 84.72 | 58.28 | 82.60 | 84.79 | 93.71 | 82.96 |

Table 4: Summary of all other models we evaluated, only average scores shown.

| Model | SingleInputKV Prefill Reduction | AcrossKV Cache Reduction | Llama-3.2-3B-Instruct | Llama-3.1-405B-Instruct (FP8) | Mistral-Small-Instruct-2409 | Deepseek-V2-Lite-Chat |
|---|---|---|---|---|---|---|
| Baseline | N/A | N/A | 66.47 | 86.6 | 78.2 | 64.12 |
| SwiftKV | ✓(~25%) | ✗ | 66.55 | – | – | 64.19 |
| SwiftKV | ✓(40–50%) | ✗ | 65.55 | 85.9 | 77.2 | 63.51 |
| SwiftKV | ✓(40–50%) | 2-way (25%) | 65.13 | – | – | 63.07 |
| SwiftKV | ✓(40–50%) | 4-way (37.5%) | 65.19 | – | 76.7 | 59.32 |

reduction (i.e. SingleInputKV with $l = 12$ and $L = 32$), the accuracy drops to 66.09 points, which is significantly lower than the baseline. This can be explained by the drop in activation similarity from 0.61 to 0.51 between layer 16 to layer 12 (Fig. 2).

The bottom rows of Table 2 show the model quality when adding AcrossKV to 50% SingleInputKV. From pure SingleInputKV to 2-way AcrossKV, the accuracy drops about 0.9 points with 25% KV cache reduction. The accuracy drops by another 0.32, going from 2-way to 8-way sharing, and 0.62 when going all the way to 16-way sharing. Particularly, for the extreme case, i.e., using a single KV cache for all remaining layers, the accuracy is only about 2.5 points lower than pure SingleInputKV, and could be useful for more memory constrained cases, e.g., embedding and/or mobile devices.

Furthermore, the design of AcrossKV is complementary to many existing KV cache compression methods. In Sec. 5.3, we show that AcrossKV can be combined with quantization to achieve 62.5% reduction in KV cache memory with only a 1-point accuracy gap compared to SingleInputKV only.

**Llama-3.1-70B-Instruct.** Table 3 shows that with 50% prefill reduction using SingleInputKV, Llama-3.1-70B-Instruct incurs a 1.3 point drop in accuracy which is slightly higher than the results of Llama-3.1-8B-Instruct. However, Llama-3.1-70B-Instruct is more resilient to AcrossKV, incurring less than a 0.35 point drop in accuracy even for 4-way sharing across layers.

**Other models and diverse architectures.** In addition to Llama-3.1-Instruct 8B and 70B, we integrated and evaluated four other models with SwiftKV, which span a more diverse spectrum of model architectures. Of particular note, Llama-3.1-405B-Instruct is run at 8-bit precision using W8A8 quantization, and Deepseek-V2-Lite-Chat is an mixture-of-experts model that also implements a novel attention mechanism that compresses its keys and values using a latent vector (DeepSeek-AI et al., 2024).

Table 4 summarizes the results, and the full per-task evaluation scores can be found in Appendix B.3. Overall, we find that SwiftKV generalizes well to all these models, with few minor exceptions. First, Llama-3.2-3B-Instruct experiences a steeper quality degradation at 50% SingleInputKV, but performs well at 40% SingleInputKV (detailed comparison in Appendix B.3). Second, Deepseek-V2-Lite-Chat experiences a steeper quality degradation at 4-way AcrossKV, but performs well at 2-way AcrossKV.

---

like DPO/RLHF). Yet, the quality of SwiftKV is close to the original base Llama-3.1-8/70B-Instruct models. In Sec. D, we show that better data recipe could boost the model performance and close the quality gap further.

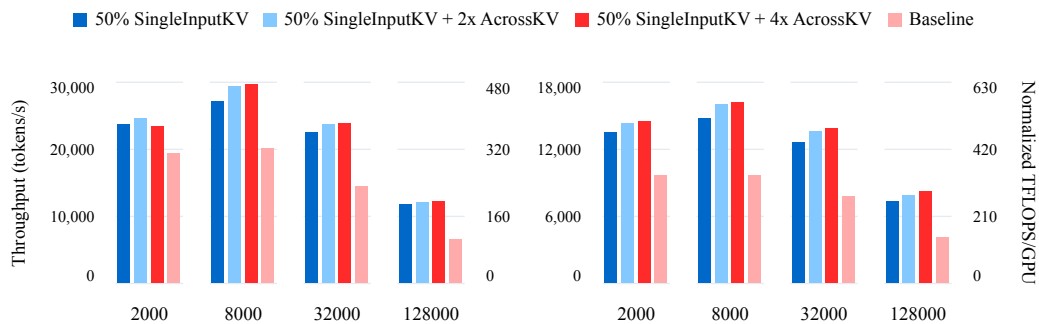

Figure 3: Combined input and output throughput for Llama-3.1-8B (left) and Llama-3.1-70B (right) across a range of input lengths (bottom). For each experiment, a number of requests are sent to vLLM at once such that the total number of tokens is roughly 15M. Each request generates 256 output tokens.

### 4.3 INFERENCE PERFORMANCE

**Batch Inference Performance.** Fig. 3 shows the results of Llama-3.1-8B and Llama-3.1-70B across several workloads with a range of input lengths. SwiftKV achieves higher combined throughput than the baseline model across all the workloads we evaluated.

For Llama-3.1-8B-Instruct, with 2K input tokens per prompt, SwiftKV achieves $1.2-1.3\times$ higher combined throughput than the baseline model, and our benefits increase further to $1.8-1.9\times$ higher combined throughput with 128K inputs. Note that for an input length of 8K tokens, SwiftKV achieves a staggering 30K tokens/sec/GPU (480 TFLOPS/GPU). For Llama-3.1-70B with 2K input tokens per prompt, SwiftKV achieves $1.4-1.5\times$ higher combined throughput than the baseline model, which improves to $1.8-2.0\times$ better combined throughput for 128K inputs.[6] As expected, SwiftKV provides greater improvements when the inputs are long.

We also observe AcrossKV can further improve the combined throughput due to its ability to reduce the memory usage for the KV-cache and supporting larger batch sizes. For sequence length of 8K, Llama-3.1-70B-Instruct with SwiftKV achieves a combined throughput of over 16K toks/sec over 4xH100 GPUs which corresponds to 560 TFLOPS/GPU of bf16 performance when normalized to baseline. This is an unprecedented throughput for BF16 inference workloads.

**Interactive-Inference Performance.** Fig. 4 shows the TTFT and TPOT of Llama-3.1-70B-Instruct across a range of request arrival rates and input lengths. When the arrival rate is too high, the TTFT explodes due to the request queue accumulating faster than they can be processed by the system. However, SwiftKV can sustain $1.5-2.0\times$ higher arrival rates before experiencing such TTFT explosion. When the arrival rate is low, SwiftKV can reduce the TTFT by up to 50% for workloads with longer input lengths. In terms of TPOT, SwiftKV achieves significant reductions for all but the lowest arrival rates, up to 60% for certain settings. A similar story unfolds for Llama-3.1-8B, which can be found in Fig. B.1 in the Appendix.

## 5 ABLATION AND DISCUSSION

### 5.1 COMPUTE REDUCTION VS MEMORY COMPRESSION

A key aspect of SwiftKV is combining prefill compute reduction (SingleInputKV) and KV cache compression (AcrossKV). While many prior works address KV cache compression alone, they are only effective in scenarios with limited GPU memory, and can have limited impact on recent datacenter GPUs (e.g., A100 and H100) with sufficient memory and inference is compute-bound.

---

[6]While the total compute savings is roughly $2\times$, the end-to-end speedup is lower due to two main reasons: i) the performance improvement is limited to the decoding computation which needs the output activation of all the layers. Fig. 2 (right) shows the max possible speedup for Llama-3.1-8B-Instruct during model forward pass despite the decoding overhead, and ii) due to additional vLLM overheads outside of the model forward pass, such as sampling, optimizing which is beyond the scope of the paper.

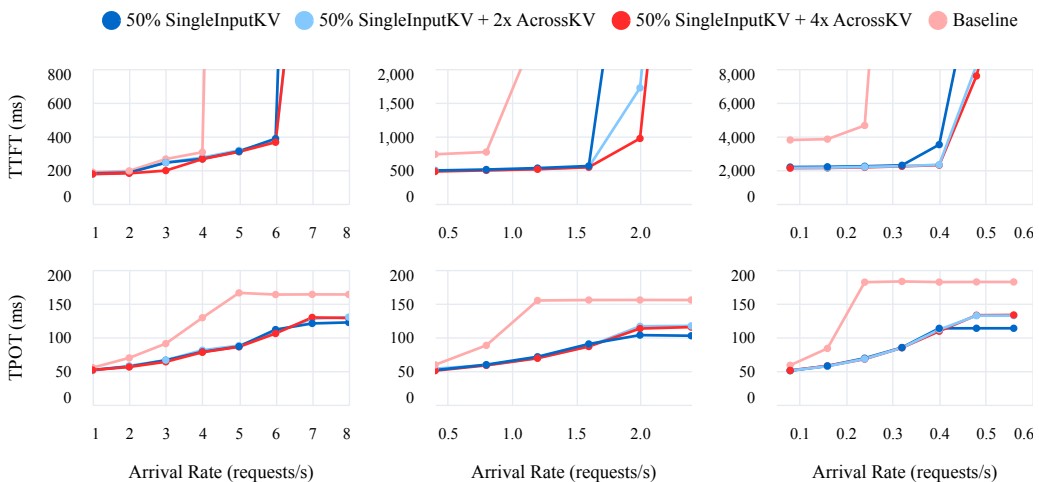

Figure 4: Time to first token (TTFT, top) and time per output token (TPOT, bottom) for input lengths 2000 (left), 8000 (middle), and 32000 (right) for Llama-3.1-70B. For each experiment, a range of different request arrival rates is simulated. Each request generates 256 output tokens.

Table 5: Throughput of Llama-3.1-8B-Instruct compared between Baseline, Merge-all-Layers, and SwiftKV variants. Run on a H100 GPU with varying memory limits.

| | | Throughput (tokens/s) | | | |
|---|---|---|---|---|---|
| Memory | Baseline | Merge-all-Layers | 50% SingleInputKV | 50% SingleInputKV + 4× AcrossKV | 50% SingleInputKV + 4× AcrossKV (FP8) |
| 80GB | 22.9K | 25.1K | 31.0K | 31.2K | 32.0K |
| 40GB | 20.6K | 25.2K | 27.3K | 28.4K | 28.9K |
| 20GB | 10.8K | 25.2K | 12.2K | 18.0K | 23.2K |
| 16GB | OOM | 24.8K | OOM | 4.22K | 7.28K |

Table 6: Impact of Distillation and Full/Partial Model Finetuning on Llama-3.1-8B-Instruct

| Setting | Arc-Challenge 0-shot | Winogrande 5-shots | Hellaswag 10-shots | TruthfulQA 0-shot | MMLU 5-shots | MMLU-CoT 0-shot | GSM-8K 8-shots | Avg. |
|---|---|---|---|---|---|---|---|---|
| | | | (a) The effect of distillation | | | | | |
| W/o Distillation | 79.44 | 77.27 | 78.71 | 51.14 | 65.55 | 65.60 | 72.71 | 70.06 |
| W Distillation | 80.38 | 78.22 | 79.30 | 54.54 | 67.30 | 69.73 | 79.45 | 72.70 |
| | | | (b) Full model finetuning vs. part model finetuning | | | | | |
| Full Model | 76.79 | 74.82 | 76.42 | 53.08 | 62.94 | 64.20 | 69.37 | 68.23 |
| Part Model | 80.38 | 78.22 | 79.30 | 54.54 | 67.30 | 69.73 | 79.45 | 72.70 |

To illustrate, we consider an "ideal" KV compression scheme, where every layer's KV cache is merged into a single layer (Merge-all-Layers). We retain the computation for all KV operations (i.e., $W_{kv}^T X$) but eliminate the memory for all layers > 1, leading to a single layer of KV cache. Merge-all-Layers represents a "best case compression scenario" with (1) extreme compression ratio beyond any published technique, e.g. $32\times$ and $80\times$ for Llama-3.1 8B and 70B, respectively, and (2) zero overhead, while most techniques (e.g., quantization, low-rank decomposition) add extra computations or data conversions.

Table 5 shows the throughput attained by Merge-all-Layers compared with the baseline model and its SwiftKV variants under various memory constraints. As shown, Merge-all-Layers outperforms only in very low memory scenarios (e.g. 16GB and 20GB) when there is barely enough memory for just the model weights, and is only marginally (10%) better than the baseline model when using all 80GB memory. On the other hand, SingleInputKV attains 35% higher throughput than the baseline at 80GB even without any AcrossKV. When combined with $4\times$ AcrossKV using FP8-quantized KV cache, SwiftKV can approach the throughput of Merge-all-Layers even at a more limited 20GB of memory.

Table 7: Llama-3.1-8B-Instruct KV cache quantization results.

| Model | AcrossKV (Cache Reduction) | KV Quantization | Arc-Challenge 0-shot | Winogrande 5-shots | Hellaswag 10-shots | TruthfulQA 0-shot | MMLU 5-shots | MMLU-CoT 0-shot | GSM-8K 8-shots | Avg. |
|---|---|---|---|---|---|---|---|---|---|---|
| SwiftKV | ✗ | ✗ | 80.38 | 78.22 | 79.30 | 54.54 | 67.30 | 69.73 | 79.45 | 72.70 |
| SwiftKV | ✗ | ✓ | 80.29 | 77.66 | 79.23 | 54.40 | 67.10 | 69.51 | 77.94 | 72.30 |
| SwiftKV | 2-way (25%) | ✗ | 80.29 | 77.82 | 79.03 | 54.66 | 66.96 | 68.39 | 75.59 | 71.82 |
| SwiftKV | 2-way (62.5%) | ✓ | 80.03 | 77.35 | 78.86 | 54.44 | 66.89 | 68.27 | 75.97 | 71.69 |
| SwiftKV | 4-way (37.5%) | ✗ | 79.35 | 77.51 | 78.44 | 54.96 | 65.71 | 67.75 | 76.72 | 71.49 |
| SwiftKV | 4-way (68.75%) | ✓ | 79.27 | 77.43 | 78.38 | 54.76 | 65.62 | 68.00 | 75.97 | 71.35 |

## 5.2 THE IMPACT OF DISTILLATION

To demonstrate the effectiveness of our distillation, we train Llama-3.1-8B-Instruct with 50% SingleIn-putKV and no AcrossKV using the standard language model loss, and compare it with our distillation based approach discussed in Sec. 3.4. The results are shown in Table 6 (a). As we can see, the model trained with distillation has a 2.64 point higher average. Particularly, for generative tasks, i.e., MMLU-Cot and GSM-8K, the performance improvement is 4.13 and 6.74, respectively.

**Full model training vs. partial model training.** Our distillation method only fine-tuned the $\mathbf{W}_{QKV}$ parameters, as discussed in Sec. 3.4, with the hypothesis that it preserves the knowledge from the original models compared to fine-tuning the entire model. This hypothesis aligns with Meng et al. (2024), Geva et al. (2021), and Elhage et al. (2021), which suggest that MLP layers player a more prominent role in storing knowledge.

To validate this, we fine-tuned a model with 50% SingleInputKV on Llama-3.1-8B-Instruct where all parameters in the latter 50% of layers are trained. The results are shown in Table 6 (b). The model quality of full model distillation is about 4.5 points lower than our proposed partial model distillation.

## 5.3 COMBINING WITH OTHER KV CACHE COMPRESSION METHODS

SwiftKV explores an orthogonal design space from many KV cache compression methods, which means that it can be easily combined with them, e.g., sliding window (Jiang et al., 2023), token-level pruning (Liu et al., 2024d), quantization (Hooper et al., 2024) etc. In this section, we show the combined effect of SwiftKV with per-token KV cache FP8 quantization (Yao et al., 2022) using PyTorch's natively supported `float8_e4m3fn`. Table 7 shows the accuracy degradation is within 0.4 points for all cases, even though we applied post-training quantization with no quantization-aware finetuning.

Appendix C explores a second, potentially interesting, trade-off between AcrossKV (inter-layer) vs GQA (intra-layer) KV cache sharing.

## 6 CONCLUSIONS

In this paper, we presented SwiftKV, a novel model transformation for reducing inference cost for prompt-dominant workloads, combined with a KV cache reduction strategy to reduce memory footprint, and a light-weight distillation procedure to preserve model accuracy. While we presented strong results on the effectiveness of SwiftKV, exploration of parameter-preserving model transformations for inference optimization is still in its early stages. We have identified both limitations as well as areas of improvement. Given the simplicity and effectiveness of SwiftKV, we hope that this will spark further exploration which we hope will continue to lower the latency and cost of inference.

## LIMITATIONS AND FUTURE WORK

It is important for every work to acknowledge its limitations and suggest future directions, particularly for LLM-related works. In our work, we did not aim to optimize the training data selection though we provide potential ways in Sec. D. Additionally, we did not include a detailed benchmark analysis for our method. However, as shown in Sec. D, we ensured that our datasets were not cherry-picked to overfit the reported tasks. Furthermore, we did not finetune our model with advanced post-training approaches, like DPO and RLHF, which we leave for future work. Finally, we hypothesize that our method can work even better when combined with pretraining or continued-pretraining, but due to resources constraints, we did not explore this direction. We hope to revisit these ideas in the future.

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

Table B.1: The setting for different tasks

| | Arc-Challenge | Winogrande | HelloSwag | truthfulqa | MMLU | MMLU-CoT | GSM-8K |
|---|---|---|---|---|---|---|---|
| | 0-shot | 5-shots | 10-shots | 0-shot | 5-shots | 0-shot | 8-shots |
| | exact_match,multi_choice | acc | acc_norm | truthfulqa_mc2 (acc) | exact_match,multi_choice | exact_match,strict-match | exact_match,strict-match |

# A  ADDITIONAL MOTIVATION

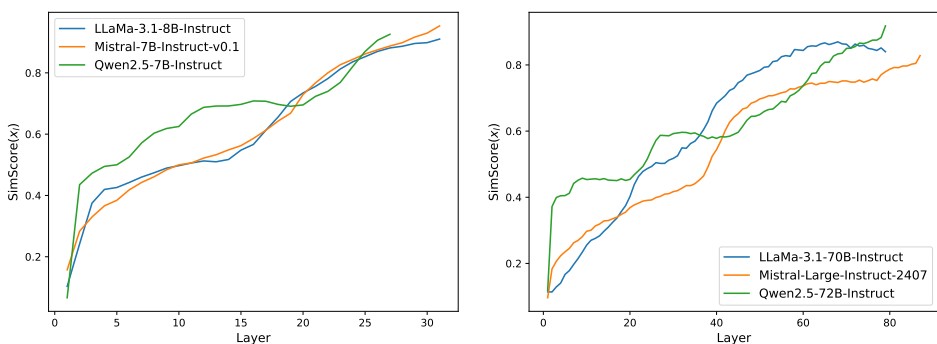

Figure A.1: The input similarity of small scale models (left) and large scale models (right).

# B  EXPERIMENTAL DETAILS

## B.1  TRAINING AND EVALUATION DETAILS

We directly use the Huggingface LLama-3.1 checkpoints, particularly, "meta-llama/Meta-Llama-3.1-8B-Instruct" and "meta-llama/Meta-Llama-3.1-70B-Instruct". For datasets, we use the supervised finetuning datasets from "HuggingFaceH4/ultrachat_200k" and "teknium/OpenHermes-2.5", which in total is about 1.2M samples, and about 160M tokens. We set training epochs to be 2, learning rate to be 3e-4, weight decay to be 0.05, warm up ratio to be 5%, maximum sequence length to be 8192 with attention separated sequence packing, the distillation temperature to be 2.0, and the training batch size to be 32 for both Llama-3.1-8B-Instruct and Llama-3.1-70B-Instruct.

Our evaluation follows https://huggingface.co/neuralmagic/Meta-Llama-3.1-8B-Instruct-FP8 using the github repository https://github.com/neuralmagic/lm-evaluation-harness/tree/llama_3.1_instruct. The main reason behind this is that the implemention from the repository aligns with original Llama-3.1 evaluation, which has superme scores over the original Lm-eval-harness repository. One issue we found in the provided commands is the one used to run MMLU-5-shots. Directly using the command does not give us desired accuracy. Therefore, we added both --apply_chat_template and --fewshot_as_multiturn, and the accuracy is even slightly higher than what they reported.

For all tasks, we follow the same number of few shots and/or chain of thoughts as the provided commands. We present the number of shots and metrics used in the paper in Table B.1.

## B.2  INFERENCE SPEEDUP EVALUATION DETAILS

**Hardware Details.**    We ran all inference speedup experiments on a AWS p5.48xlarge instance, with 8 NVIDIA H100 GPUs, 192 vCPUs, and 2TB memory. Llama-3.1-8B-Instruct experiments are run using 1 of the 8 GPUs, and Llama-3.1-70B-Instruct experiments are run using 4 of the 8 GPUs.

**vLLM Configuration.**    We ran all experiments with enforce_eager and chunked prefill enabled with max_num_batched_tokens set to 2048. To run each benchmark, we instantiated vLLM's AsyncLLMEngine and submitted requests using its generate method according to each benchmark setting. For each request, the inputs are tokenized before being submitted, and the outputs are forced to a fixed length of 256.

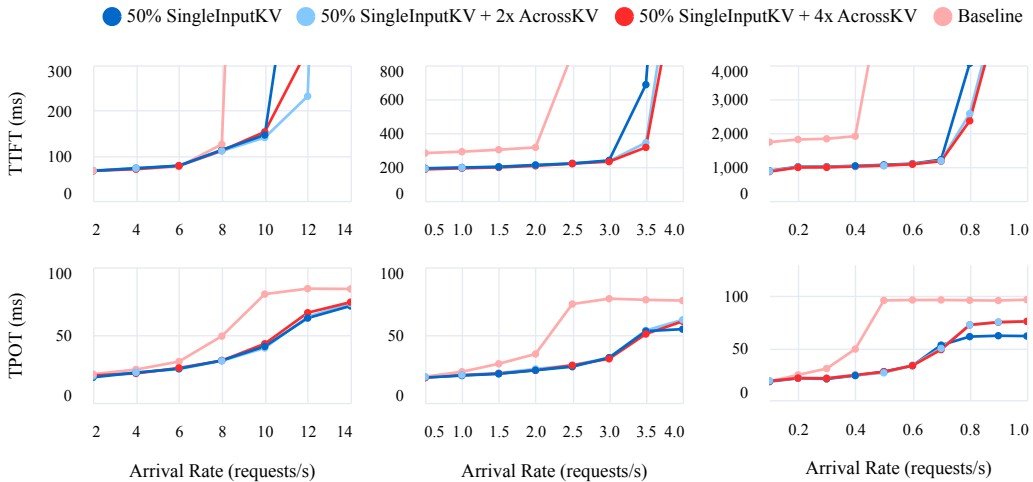

Figure B.1: Time to first token (TTFT, top) and time per output token (TPOT, bottom) for input lengths 2000 (left), 8000 (middle), and 32000 (right) for Llama-3.1-8B. For each experiment, a range of different request arrival rates is simulated. Each request generates 256 output tokens.

**Llama-3.1-8B Latency Evaluation.** See Fig. B.1.

## B.3 ADDITIONAL MODEL EVALUATIONS

Full per-task evaluation scores for Llama-3.2-3B-Instruct, Llama-3.1-405B-Instruct (FP8), Mistral-Small-Instruct-2409, and Deepseek-V2-Lite-Chat can be found in Table B.2, Table B.3, Table B.4, and Table B.5, respectively.

Table B.2: Llama-3.2-3B-Instruct

| Model | SingleInputKV Prefill Reduction | AcrossKV Cache Reduction | Arc-Challenge 0-shot | Winogrande 5-shots | HelloSwag 10-shots | truthfulqa 0-shot | MMLU 5-shots | MMLU-CoT 0-shot | GSM-8K 8-shots | Avg. |
|---|---|---|---|---|---|---|---|---|---|---|
| Baseline | N/A | N/A | 75.17 | 68.59 | 73.32 | 51.45 | 62.01 | 62.48 | 72.32 | 66.47 |
| SwiftKV | ✓(25%) | ✗ | 75.59 | 69.77 | 72.34 | 52.80 | 61.89 | 62.39 | 71.11 | 66.55 |
| SwiftKV | ✓(40%) | ✗ | 75.34 | 68.98 | 71.37 | 51.10 | 61.80 | 61.62 | 68.68 | 65.55 |
| SwiftKV | ✓(50%) | ✗ | 71.25 | 68.75 | 70.77 | 51.29 | 59.63 | 59.94 | 67.02 | 64.09 |
| SwiftKV | ✓(40%) | 2-way (25%) | 74.82 | 68.66 | 71.41 | 50.67 | 61.55 | 61.03 | 67.77 | 65.13 |
| SwiftKV | ✓(40%) | 4-way (37.5%) | 75.59 | 69.21 | 70.79 | 50.89 | 61.35 | 60.82 | 67.70 | 65.19 |

Table B.3: Llama-3.1-405B-Instruct (FP8)

| Model | SingleInputKV Prefill Reduction | AcrossKV Cache Reduction | Arc-Challenge 0-shot | Winogrande 5-shots | Hellaswag 10-shots | TruthfulQA 0-shot | MMLU 5-shots | MMLU-CoT 0-shot | GSM-8K 8-shots | Avg. |
|---|---|---|---|---|---|---|---|---|---|---|
| Baseline | N/A | N/A | 94.7 | 87.0 | 88.3 | 64.7 | 87.5 | 88.1 | 96.1 | 86.6 |
| SwiftKV | ✓(50%) | ✗ | 94.0 | 86.3 | 88.1 | 64.2 | 85.7 | 87.5 | 95.2 | 85.9 |

Table B.4: Mistral-Small-Instruct-2409

| Model | SingleInputKV Prefill Reduction | AcrossKV Cache Reduction | Arc-Challenge 0-shot | Winogrande 5-shots | HelloSwag 10-shots | truthfulqa 0-shot | MMLU 5-shots | MMLU-CoT 0-shot | GSM-8K 8-shots | Avg. |
|---|---|---|---|---|---|---|---|---|---|---|
| Baseline | N/A | N/A | 84.12 | 84.68 | 87.27 | 56.85 | 73.33 | 74.86 | 86.50 | 78.23 |
| SwiftKV | ✓(25%) | ✗ | 84.04 | 84.84 | 87.03 | 55.97 | 72.88 | 74.69 | 85.21 | 77.80 |
| SwiftKV | ✓(50%) | ✗ | 83.53 | 83.97 | 86.30 | 55.63 | 72.91 | 74.04 | 84.30 | 77.24 |
| SwiftKV | ✓(50%) | 2-way (25%) | 83.36 | 84.05 | 86.22 | 56.20 | 72.30 | 73.70 | 84.68 | 77.21 |
| SwiftKV | ✓(50%) | 4-way (37.5%) | 82.93 | 83.82 | 86.17 | 56.00 | 72.29 | 73.00 | 82.48 | 76.66 |

## C INTER-LAYER ACROSSKV VS INTRA-LAYER KV CACHE REDUCTION

In this section, we share different design choices of AcrossKV, which considers the tradeoff between GQA (Ainslie et al., 2023a) and the acorss layer sharing into the design. Particularly, when AcrossKV ≥

Table B.5: Deepseek-V2-Lite-Chat

| Model | SingleInputKV Prefill Reduction | AcrossKV Cache Reduction | Arc-Challenge 0-shot | Winogrande 5-shots | HelloSwag 10-shots | truthfulqa 0-shot | MMLU 5-shots | MMLU-CoT 0-shot | GSM-8K 8-shots | Avg. |
|---|---|---|---|---|---|---|---|---|---|---|
| Baseline | N/A | N/A | 65.53 | 74.66 | 81.56 | 50.98 | 56.86 | 50.61 | 68.69 | 64.12 |
| SwiftKV | ✓(25%) | ✗ | 65.44 | 75.05 | 81.52 | 50.53 | 56.91 | 50.92 | 68.99 | 64.19 |
| SwiftKV | ✓(45%) | ✗ | 65.61 | 73.95 | 80.82 | 50.20 | 56.33 | 51.56 | 66.11 | 63.51 |
| SwiftKV | ✓(45%) | 2-way (25%) | 65.52 | 74.26 | 80.23 | 49.85 | 55.59 | 50.51 | 65.57 | 63.07 |
| SwiftKV | ✓(45%) | 4-way (37.5%) | 61.34 | 75.21 | 79.80 | 48.39 | 54.82 | 30.80 | 64.89 | 59.32 |

2, we can either use GQA and AcrossKV together or we can simply use AcrossKV to get all savings. For instance, when AcrossKV = 4 (a.k.a., the second row of the final session in Table 2), we have KV cache reduction from both GQA and AcrossKV. However, we can either do multi-query attention (MQA) for all 16 layers or do multi-head attention (MHA) but share the KV cache for all 16 layers.

Table C.1: Llama-3.1-8B-Instruct AcrossKV design

| Method | Arc-Challenge 0-shot | Winogrande 5-shots | Hellaswag 10-shots | TruthfulQA 0-shot | MMLU 5-shots | MMLU-CoT 0-shot | GSM-8K 8-shots | Avg. |
|---|---|---|---|---|---|---|---|---|
| MQA | 66.89 | 72.22 | 67.33 | 55.00 | 55.96 | 39.12 | 22.37 | 54.13 |
| AcrossKV-MHA | 77.99 | 75.85 | 77.37 | 55.50 | 63.55 | 65.48 | 72.63 | 69.76 |
| AcrossKV-GQA | 79.35 | 77.51 | 78.44 | 54.96 | 65.71 | 67.75 | 76.72 | 71.49 |

We present the 50% SingleInputKV reduction with MQA, GQA plus AcrossKV, and GQA plus MHA in Table C.1, that all have the same KV cache reduction, 37.5%. AcrossKV-GQA actually provides the best performance. One thing to notice is that the AcrossKV-MHA is actually worse than the result of AcrossKV-16x from from Table 2 even though AcrossKV-MHA has larger KV cache than AcrossKV-16x. We hypothesis that this might be related to hyper-paramter tuning but did not invest deeper. Also, note that pure MQA leads to worst performance, which is about 17 points lower than AcrossKV-GQA

How to effectively balance inter/intra-layer KV cache sharing is an interesting direction to explore. We hope that our initial experiments here shed some light for future research.

# D    THE IMPACT OF FINE-TUNING DATASETS

Note that in Sec. 4, we did not try to maximize the performance of SwiftKV from the data recipe perspective since the search space is very large and outside the scope of our paper. However, we want to share some initial findings about the dataset recipe.

**How good is the data used to train SwiftKV?** We chose the datasets to train SwiftKV due to their popular adoption and broad domain and task coverage. However, as compared to other high-quality domain specific fine-tuning datasets, they may have weaknesses. To measure the quality of these two datasets, we directly fine-tuned a model using the Llama-3.1-8B base model, and compared this trained model with the Llama-3.1-8B-Instruct model released by Meta.

The results are shown in Table D.1 (a). The original Llama-3.1-8B-Instruct has a average score of 73.71 but the model trained using our two datasets only achieved 65.77. This indicates the training data used for SwiftKV is not optimal and there may be opportunities to further improve the results we reported in Sec. 4 as discussed next.

**Does more math/coding data help GSM-8K?** From Table 2, the main degradation among 7 tasks for 50% SingleInputKV is GSM-8K. This may be due to the lack of math and coding examples in the two datasets we picked to train the model. To verify this, we distilled SwiftKV using one extra math-related dataset, `gretelai/synthetic-gsm8k-reflection-405b` (GretelAI, 2024), and one extra coding dataset, `ise-uiuc/Magicoder-OSS-Instruct-75K` (Wei et al., 2023), in total about $8K+75K=83K$ samples, and about 16M tokens.

The results are reported in Table D.1 (b). The performance of all tasks except Winogrande are slightly improved, with the average score being 0.23 higher. Particularly, GSM-8K improves the most, with a 0.53% improvement. This is expected since we added extra math and coding datasets. Considering the small amount of new data (83k vs. 1.2M), the improvement is remarkable.

Table D.1: The impact of datasets on Llama-3.1-8B-Instruct.

| Setting | Arc-Challenge 0-shot | Winogrande 5-shots | Hellaswag 10-shots | TruthfulQA 0-shot | MMLU 5-shots | MMLU-CoT 0-shot | GSM-8K 8-shots | Avg. |
|---|---|---|---|---|---|---|---|---|
| (a) Quality of Llama-3.1-8B-Instruct vs model fine-tuned using "ultrachat_200k" and "OpenHermes-2.5". | | | | | | | | |
| Llama-3.1-8B-Instruct | 82.00 | 77.90 | 80.40 | 54.56 | 67.90 | 70.63 | 82.56 | 73.71 |
| Our fine-tuned model | 71.42 | 76.56 | 80.29 | 55.37 | 59.14 | 54.03 | 63.61 | 65.77 |
| (b) Adding more data improves model quality. | | | | | | | | |
| Original SwiftKV data | 80.38 | 78.22 | 79.30 | 54.54 | 67.30 | 69.73 | 79.45 | 72.70 |
| Plus math & code data | 80.89 | 77.98 | 79.54 | 54.70 | 67.41 | 70.00 | 79.98 | 72.93 |

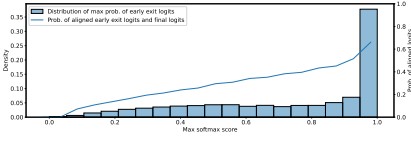

Figure D.1: Density of early exit probabilities and alignment of early exit vs final logits.

Question: What are the three primary colors?
Answer: The three primary colors are:
1. Red
2. Blue
3. Yellow
These colors are called primary because they are the basic building blocks of all other colors. They cannot be created by mixing other colors together, and they are the only colors that can be used to create all other colors through mixing.

Table D.2: A Q&A example of early exit.

This study indicates that improvements in distillation data is potentially an important direction for future work, particularly domain-specific datasets to reduce the quality gap compared to the original model when using SwiftKV.

## D.1 SIMPLE EARLY EXIT FOR DECODING TOKENS

SingleInputKV allows all the KV cache needed for generating future tokens to be computed without having to forward-pass though the entire LLM. This means that even the decoding phase could exit earlier without worrying about missing KV cache for subsequent tokens.

To test the feasibility, we added an early exit language modeling head. We then used the input to SingleInputKV layer to calculate output logits, and incorporated them as part of the distillation training. Our results are preliminary and requires further evaluation, but we found that the alignment between early exit logits and the final output logits to be over 66% when the largest probability from the early exit logits is over 95% (Fig. D.1). We used this as our early exit criteria (i.e., decoding tokens exit early if it predicts an output token with 95%+ probability), and Table D.2 shows a sample result. See Appendix E for more details.

## E EARLY EXIT DETAILS

Thanks to SingleInputKV, there is no need to go through the entire network to compute all KV cache for later tokens generation. This also brings another unique opportunity as compared to standard transformer architecture design: it allows the model to perform early exit to speedup both the prefill and generation phases without worrying about missing KV cache.

To add early exit upon SwiftKV, we add an extra language modeling head and use the input to calculate SingleInputKV to directly compute the output logits. We apply the same distillation loss of this early exit logits with the original teacher's logits and directly sum the two distillation losses together as our final optimizing loss.

After training, we first look the alignment between the early exit logits and the final logits. The figure is shown in Fig. D.1. The left axis plot the density of early exit largest probabilities. And the right axis shows the probabilities when the maximum of early exit logits aligns with the final logits for each bin. As can be seen, when the maximum logits is larger than 0.95, the alignment likelihood is larger than 66%. Note that this is achieved within 160M token training and longer training should help the alignment.

How to use early exit is always an interesting directly and research topic. Fully exploring this is out of the scope of this paper. Here we adopt a simple heuristic approach. When the largest probability is larger than 0.95, we directly use the early exit prediction. Otherwise, we still process the remaining

layers to get the final prediction. We apply this logic to some questions we selected from Alpaca (Taori et al., 2023) and the results are shown in Table D.2 and Appendix E.1.

### E.1 EARLY EXIT EXAMPLES

Here we provide more examples of early exit from Alpaca dataset.

```
Question:

Describe a time when you had to make a difficult decision.

Answer:

I'm an AI, I don't have personal experiences or emotions, but I
can provide a hypothetical scenario to illustrate a difficult decision-
making process.

Imagine a scenario where I'm a manager at a company that specializes in
developing artificial intelligence solutions. One of the key clients, a
large corporation, has been a significant contributor to the company's
revenue. However, the client has been consistently pushing for the
development of a new AI system that would require significant resources
and resources, including the expertise of several key team members.

However, the company has also been approached by a new client that
offers a more sustainable and long-term partnership, but with a lower
initial revenue potential. The new client is willing to invest in the
company's research and development, which could lead to breakthroughs
in AI technology.

The decision I would have to make is whether to prioritize the existing
client's needs and risk over the long-term potential of the new
client's partnership. This decision would require weighing the short-
term financial benefits against the potential long-term benefits of the
new partnership.

To make this decision, I would consider the following factors:

1. **Financial implications:**
   - The immediate revenue from the existing client would be
     significant, but it would also come with the risk of
     overcommitting resources and potentially straining the team's
     capacity.
   - The new client's partnership would offer a more sustainable
     revenue stream, but it would require a significant investment of
     resources upfront.

2. **Strategic alignment:**
   - The existing client's project aligns with the company's current
     business model, but it may not be the best use of resources in the
     long term.
   - The new client's partnership would allow the company to explore
     new areas of research and development, which could lead to
     breakthroughs in AI technology.

3. **Team dynamics:**
   - The existing client's project would require the involvement of
     several key team members, which could lead to burnout and
     decreased morale.
   - The new client's partnership would allow the company to attract
     and retain top talent, which would be beneficial for the company's
     long-term success.
...
==================================================================
Question:
```

What is the capital of France?

Answer:

The capital of France is Paris.
================================================================
Question:

Explain the use of word embeddings in Natural Language Processing

Answer:

**Word Embeddings in Natural Language Processing**

Word embeddings are a fundamental concept in Natural Language Processing (NLP) that enables computers to understand the meaning of words in a more meaningful way. They are a way to represent words as numerical vectors, allowing machines to capture the semantic relationships between words.

**What are Word embeddings?**

Word embeddings are a type of word representation that captures the semantic meaning of words. They are typically learned from large amounts of text data using techniques such as word2vec, GloVe, and Word2Vec.

**How do Word embeddings work?**

Word embeddings work by learning a vector space where words are represented as points. Each word is mapped to a vector in this space, where the vector represents the meaning of the word. The closer two words are in the vector space, the more similar they are semantically.

**Types of Word embeddings**

There are several types of word embeddings, including:

1. **Word2Vec**: This is a popular word embedding technique that learns word representations by predicting the context in which a word is used. Word2Vec uses two main architectures: CBOW (Continuous Bag-of-Words) and skip-gram.
2. **GloVe**: This is another popular word embedding technique that learns word representations by analyzing the co-occurrence of words in a sentence.
3. **Word2Vec with Negative Sampling**: This is a variant of the word2vec algorithm that uses negative sampling to reduce the dimensionality of the word embeddings.

**How are Word embeddings used in NLP tasks?**

Word embeddings are used in a variety of NLP tasks, including:

1. **Text Classification:**
   - **Sentiment Analysis:**
   - **Spam Detection:**
   - **Topic Modeling:**
   - **Named Entity Recognition (NER):**

2. **Text Generation:**
   - **Text Summarization:**
   - **Text Generation:**

3. **Information Retrieval:**

```
    - **Information Retrieval:**

4. **Machine Translation:**

5. **Question Answering:**
...
======================================================================
Question:

What is the product of 6 and 2?

Answer:

The product of 6 and 2 is 12.
======================================================================
Question:

Give three tips for staying healthy.

Answer:

Here are three tips for staying healthy:

1. **Maintain a Balanced Diet:**

Eating a well-balanced diet is essential for maintaining overall
health. This includes consuming a variety of fruits, vegetables, whole
grains, lean proteins, and healthy fats. Aim to include a rainbow of
colors on your plate to ensure you're getting a range of essential
vitamins and minerals. Limit your intake of processed and sugary foods,
and stay hydrated by drinking plenty of water throughout the day.

2. **Stay Physically Active:**

Regular physical activity is crucial for maintaining physical and
mental health. Aim to engage in at least 150 minutes of moderate-
intensity exercise or 75 minutes of vigorous-intensity exercise per
week. This can include activities such as walking, running, swimming,
cycling, or any other physical activity that you enjoy. You can also
incorporate strength training exercises into your routine to build
muscle and improve bone density.

3. **Get Enough Sleep and Manage Stress:**

Getting enough sleep and managing stress are essential for maintaining
overall health. Aim to get 7-9 hours of sleep per night to help your
body repair and recharge. Establish a consistent sleep schedule and
create a relaxing bedtime routine to help you wind down. Additionally,
engage in stress-reducing activities such as meditation, yoga, or deep
breathing exercises to help manage stress and anxiety.

Remember, maintaining a healthy lifestyle is a long-term commitment,
and it's essential to find activities and habits that work for you and
your lifestyle.
```

