# OpenReview forum: "SwiftKV: Fast Prefill-Optimized Inference with Knowledge-Preserving Model Transformation"
_ICLR.cc/2025/Conference — Submitted to ICLR 2025_

### Official Review · Reviewer_2qP6 · 2024-10-29

**Soundness:** 2
**Presentation:** 3
**Contribution:** 2
**Rating:** 5
**Confidence:** 4

**Summary:**

This paper proposes SwiftKV, a method that reduces LLM inference latency while preserving knowledge. SwiftKV combines Early Exit, KV cache compression, and Knowledge Distillation techniques, demonstrating latency improvements in performance evaluation.

**Strengths:**

1. The writing is good.
2. Experimental results indicate that the proposed method effectively reduces latency while preserving knowledge.

**Weaknesses:**

1. Limited Efficient Experiments: VLLM serves as the only baseline in the performance results, which limits the demonstration of this work’s necessity and effectiveness. The authors’ method is a lossy optimization approach, and they should compare it with more serving systems to demonstrate respective performance improvements and knowledge retention. Although other methods may not conflict with the authors' approach, they may not be easily integrated (e.g. the strategy of Early Exit can hardly apply to Speculative Decoding[1], or combined with certain sparse attention methods like PowerInfer[2], quantization method like GPTQ[3] may result in significant performance degradation.). If the authors cannot demonstrate the effectiveness of their method compared to others, or show that it can integrate with other methods for added benefits, the significance of this work is greatly diminished.

2. Lack of Key Assumptions: Some critical assumptions are missing, such as noting that latency-sensitive servers often adopt disaggregated systems to handle the prefill and decode stages separately. This omission could impact the reported TTFT and TPOT performance results, because in the disaggregated systems, TPOT will hardly be influenced due to improvements in the prefill stage.

[1] Cai, Tianle, et al. "Medusa: Simple llm inference acceleration framework with multiple decoding heads." arXiv preprint arXiv:2401.10774 (2024).

[2] Song, Yixin, Zeyu Mi, Haotong Xie, and Haibo Chen. "Powerinfer: Fast large language model serving with a consumer-grade gpu." arXiv preprint arXiv:2312.12456 (2023).

[3] Frantar, Elias, Saleh Ashkboos, Torsten Hoefler, and Dan Alistarh. "Gptq: Accurate post-training quantization for generative pre-trained transformers." arXiv preprint arXiv:2210.17323 (2022).

**Questions:**

1. The code link appears to be invalid. Could you make the code open-source to enhance reproducibility?
2. SwiftKV focuses primarily on optimization during the prefill stage. How should we interpret the decrease in TPOT shown in the performance results?
3. Could you provide results comparing the performance of SwiftKV with more competitive baselines, such as Minicache, as mentioned in your paper? Could you clarify the connections and differences between your method and existing work, including its strengths and weaknesses? Could you demonstrate whether your method can be integrated with other approaches? Additionally, could you outline the potential application scenarios for your method?
4. As I understand, most datasets used in your paper consist of multiple-choice questions, leading to longer prefill times and shorter decoding times. I’m interested in seeing SwiftKV's performance on more diverse datasets.

---

> ### Author Response · Authors · 2024-11-19
> **Response to Reviewer 2qP6**
>
> Thank you for your detailed review of our paper and for your feedback.
>
> **On more serving systems and lossy optimization:** We clarify that vLLM is not a comparison in our study but it is used as our inference backend, as vLLM is the most widely used inference engine in production today. Although SwiftKV produces models with some quality loss, the quality loss does not depend on the inference system and can be evaluated independently. The knowledge retention and total compute reduction does not change across inference systems.
>
> We chose vLLM to demonstrate that it is possible to achieve further improvements over a strong baseline system. Supporting SwiftKV for vLLM was part of the contribution of the paper. Other systems do not support SwiftKV either, and would require implementation for each which is beyond the scope of our paper.
>
> **On integration with other techniques:** In Sec 5.4 and 5.5, we showed that SwiftKV can be integrated with KV cache quantization and early exit for decoding tokens. In our comment for all reviewers, we also showed results for Llama-3.1-405B-Instruct quantized with W8A8. The model quality gap for Llama-405B is smaller than the other models we tested, which indicates that our method may be more robust for larger models.
>
> While we agree that integration between SwiftKV and other techniques is important, it has a very large space of combinations and is infeasible for us to implement and evaluate. However, we shall add more substantial discussions to our paper. In short, SwiftKV is fully compatible with speculative decoding because we still execute all layers for decode tokens and can provide output logits to verify draft tokens with. For sparse attention, SwiftKV is agnostic to attention sparsity and, in principle, can be used in combination to achieve a compound reduction in total inference compute.
>
> **Comparison with pruning and early-exit:**
>
> For early-exit, one recent work is https://arxiv.org/pdf/2404.16710. However, the accuracy gap is much larger than ours even with continued pre-training on 500B tokens and/or training from scratch using 28B tokens (Figure 8 / Table 2), while our training budget is orders of magnitude smaller (160M x 2 epoch = 320M tokens). Also, even if they are able to early-exit one token, if any later token requires all layers, then all the missing KV-Cache belonging to the exited tokens still need to be computed. As such, it may speed up latency-sensitive use cases but it may not reduce the total computation.
>
> For pruning, two recent works are Llama-3.2 [[1]] and Minitron [[2]]. Although they have more substantial data curation (both use a much larger model to generate synthetic data, and both use orders of magnitude more data) than we do, our method has much better accuracy (e.g., Minitron-4B has <61% on MMLU, but ours is 67.3% for 50% SingleInputKV). On the other hand, our method only applies to prefill tokens (making it suitable for input-heavy workloads) while they apply to both prefill and decode.
>
> We will add discussions about this as well as the suggested related works in revising our paper.
>
> To answer your specific questions:
>
> 1. **Open source release:** We will certainly make the code open source after the review process and anonymity requirements are lifted. We will include the distillation code, inference code, as well as SwiftKV models.
> 2. **Improvements to TPOT:** This improvement is due to mixing prefill and decode tokens in each minibatch, which is widely adopted by inference systems today, including vLLM. The decode tokens in each minibatch also benefit from the prefill tokens being processed faster. As you correctly pointed out, this may not apply to prefill-decode disaggregation. However, the Throughput/TTFT improvements are still valid. We will add this discussion to our paper.
> 3. **Compared with KV cache reduction:** Our key contribution is prefill compute reduction (SingleInputKV) combined with KV cache compression, while prior works only address KV cache compression. We refer to our top-level comment for all reviewers for a more detailed discussion. In it, we compared 50% SingleInputKV with Merge-all-Layers, which represents an idealized version of MiniCache.
> 4. **Diverse set of datasets:** The datasets used for training the model and evaluating the model (mmlu, gsm8k etc) are not the same as the workload we use to evaluate SwiftKV’s throughput and latency. For the latter we use a traffic pattern from our production where the requests have a median of 2K+ input tokens, with about 256 output tokens. This observation that inputs are much larger than outputs in production was the motivation for SwiftKV. However, if a different traffic pattern was used where generation is larger than prefill, then SwiftKV would have limited benefits.
>
> [1]: https://ai.meta.com/blog/llama-3-2-connect-2024-vision-edge-mobile-devices/
> [2]: https://developer.nvidia.com/blog/how-to-prune-and-distill-llama-3-1-8b-to-an-nvidia-llama-3-1-minitron-4b-model/

---

> ### Comment · Reviewer_2qP6 · 2024-11-23
>
> Your response did not fully address my concerns.
>
> First, I kindly request to see the acceleration effects of this method demonstrated on other systems (not just vllm), or a comparison with the optimization effects of other systems. It would be helpful to understand whether your method can effectively integrate with optimizations from other systems or if it demonstrates superior performance compared to them.
>
> Secondly, I would like to ask for ablation studies specifically focusing on the kvcache compression component, as this would help substantiate the necessity of adopting AcrossKV in this approach. While I appreciate the additional experiments you provided, they only validated the necessity of early exit.
>
> Thirdly, the work suffers from the lack of evaluations on datasets with varying prefill and decoding characteristics to showcase the method’s practical effectiveness. In fact, the proposed approach appears primarily beneficial for optimizing the prefill stage. Therefore, the usage of TPOT might mislead readers into overestimating the contribution of this paper
>
> Therefore, I will maintain my previous rating.

---

> > ### Author Response · Authors · 2024-11-27
> > **Response to Reviewer 2qP6 (1/2)**
> >
> > We thank the reviewer for reviewing our response and clarifying the concerns. Below, we provide additional experiments on a non-vLLM system, AcrossKV, and more realistic request distributions. We shall add these additional results to our revised paper.
> >
> > **On acceleration effects of this method demonstrated on other systems:** We additionally integrated SwiftKV with SGLang (https://github.com/sgl-project/sglang) and repeated our throughput experiments, as shown below. Overall, we obtain similar percentage improvements over the baseline for SGLang as vLLM (1.4 - 1.8x higher throughput for Llama-8B, and 1.5 - 1.8x for Llama-70B).
> >
> > We would also like to note that most inference systems combine multiple system optimizations proposed in prior works. By showing SwiftKV’s performance on both vLLM and SGLang, we have also shown that SwiftKV is compatible with PagedAttention (https://arxiv.org/pdf/2309.06180), RadixAttention (https://arxiv.org/pdf/2312.07104), FlashAttention (https://arxiv.org/abs/2307.08691), FlashInfer (https://github.com/flashinfer-ai/flashinfer), and SplitFuse/Sarathi (https://arxiv.org/abs/2401.08671, https://arxiv.org/abs/2403.02310), each of which are implemented in either vLLM or SGLang.
> >
> > Llama-3.1-8B-Instruct on SGLang
> >
> > Input length | Output length | Baseline (tokens/s) | 50% SingleInputKV (tokens/s) | 50% SingleInputKV + 4x AcrossKV (tokens/s)
> > |-|-|:-:|:-:|:-:|
> > 2000 | 256 | 27.4K | 36.2K | 38.9K
> > 8000 | 256 | 22.9K | 31.0K | 34.0K
> > 32000 | 256 | 16.9K | 25.9K | 26.6K
> > 128000 | 256 | 7.66K | 13.2K | 14.0K
> >
> > Llama-3.1-70B-Instruct on SGLang
> >
> > Input length | Output length | Baseline (tokens/s) | 50% SingleInputKV (tokens/s) | 50% SingleInputKV + 4x AcrossKV (tokens/s)
> > |-|-|:-:|:-:|:-:|
> > 2000 | 256 | 11.6K | 15.7K | 17.3K
> > 8000 | 256 | 10.8K | 16.1K | 17.8K
> > 32000 | 256 | 8.82K | 14.0K | 15.3K
> > 128000 | 256 | 4.78K | 8.21K | 8.75K
> >
> > **On experiments on AcrossKV’s KV cache compression effects:** As we mentioned in our top-level comment for all reviewers, KV cache compression by itself (no matter how effective) may have limited impact in compute-bound scenarios when GPU memory is plentiful (which is our primary target). Although we do not have immediate access to memory-limited GPUs, we can simulate such scenarios by artificially limiting the memory used by the inference system.
> >
> > As demonstrated below and in our top-level comment, AcrossKV (and other memory compression techniques) have a marginal impact when the full 80GB of memory is used on each GPU. However, there are some scenarios, when the total memory is barely enough to hold the model weights alone, in which AcrossKV makes a greater impact (bolded cells in the tables below). This is because the token-concurrency in these scenarios is very low, and performance can be improved by processing more tokens in parallel, which is enabled by AcrossKV.
> >
> > Llama-3.1-8B-Instruct
> >
> > GPUs x Memory | Baseline (tokens/s) | 50% SingleInputKV (tokens/s) | 50% SingleInputKV + 4x AcrossKV (tokens/s)
> > |-|:-:|:-:|:-:|
> > 1 x 80GB | 22.9K | 31.0K | 31.2K
> > 1 x 40GB | 20.6K | 27.3K | 28.4K
> > 1 x 20GB | 10.8K | 12.2K | **18.0K**
> > 1 x 16GB | OOM | OOM | **4.22K**
> >
> > Llama-3.1-70B-Instruct
> >
> > GPUs x Memory | Baseline (tokens/s) | 50% SingleInputKV (tokens/s) | 50% SingleInputKV + 4x AcrossKV (tokens/s)
> > |-|:-:|:-:|:-:|
> > 4 x 80GB | 10.8K | 16.1K | 16.3K
> > 4 x 60GB | 10.0K | 14.6K | 16.1K
> > 4 x 40GB | 6.55K | 8.09K | **10.7K**
> > 4 x 35GB | OOM | OOM | **1.71K**

---

> > > ### Author Response · Authors · 2024-11-27
> > > **Response to Reviewer 2qP6 (2/2)**
> > >
> > > **On practical datasets with varying prefill and decoding characteristics:** We apologize that we misinterpreted the reviewer’s original question. We provide additional evaluations using the ShareGPT dataset, which consists of real-world conversations between users and ChatGPT. To better match our own observed request lengths (i.e. inputs >= 10x outputs), and to cover a broader range of scenarios, we also benchmark different versions of ShareGPT filtered by minimum input/output ratios. These datasets should preserve the internal diversity of request lengths from ShareGPT. We report the average input/output length ratios and the measured performance for each of these filtered datasets below.
> > >
> > > Overall, we observe similar percentage improvements from SwiftKV as our synthetic-dataset experiments, i.e. 1.25 - 1.7x and 1.25 - 1.8x higher throughput for Llama-8B and Llama-70B respectively for average length ratios up to ~100 (similar ratio to our 32K input length experiments). We hope this additional result adds more confidence to the practical applicability of SwiftKV on real-world dataset distributions.
> > >
> > > Llama-3.1-8B-Instruct
> > >
> > > Min length ratio filter | Avg length ratio of filtered dataset | Baseline (tokens/s) | 50% SingleInputKV (tokens/s) | 50% SingleInputKV + 4x AcrossKV (tokens/s)
> > > |-|-|:-:|:-:|:-:|
> > > 0 (Original) | 1.5 | 23.7K | 27.6K | 29.4K
> > > 0.2 | 3.4 | 25.8K | 31.3K | 31.9K
> > > 1 | 6.5 | 27.2K | 35.1K | 37.3K
> > > 2 | 10 | 30.3K | 41.5K | 43.7K
> > > 10 | 26 | 37.1K | 54.7K | 56.6K
> > > 20 | 40 | 37.7K | 57.6K | 59.9K
> > > 100 | 150 | 40.3K | 64.2K | 67.0K
> > >
> > > Llama-3.1-70B-Instruct
> > >
> > > Min length ratio filter | Avg length ratio of filtered dataset | Baseline (tokens/s) | 50% SingleInputKV (tokens/s) | 50% SingleInputKV + 4x AcrossKV (tokens/s)
> > > |-|-|:-:|:-:|:-:|
> > > 0 (Original) | 1.5 | 9.73K | 11.2K | 12.2K
> > > 0.2 | 3.4 | 10.4K | 13.2K | 14.2K
> > > 1 | 6.5 | 11.4K | 15.6K | 16.0K
> > > 2 | 10 | 12.6K | 18.0K | 19.0K
> > > 10 | 26 | 14.1K | 22.6K | 23.2K
> > > 20 | 40 | 14.1K | 22.9K | 24.1K
> > > 100 | 150 | 14.6K | 24.9K | 25.8K
> > >
> > > **On interpretation of the TPOT metric:** The majority of popular open-source inference systems today run prefill and decoding on the same hardware devices by default, either by mixing prefill/decode in the same batch or by interleaving prefill-only and decode-only batches (e.g. vLLM, SGLang, TRT-LLM). In these systems, TPOT is influenced by both prefill compute and decode compute. Therefore, we believe that SwiftKV’s improvements to TPOT are not misleading but rather an expected outcome for most users of these systems. We thank the reviewer for pointing out disaggregated inference systems and will add a discussion in our paper that clarifies when TPOT improvements should and should not be expected.

---

### Official Review · Reviewer_Zy9U · 2024-10-29

**Soundness:** 3
**Presentation:** 2
**Contribution:** 2
**Rating:** 6
**Confidence:** 3

**Summary:**

SwiftKV proposes several techniques to improve to reduce computation and memory footprint for LLM inference while maintaining a similar level of accuracy. In particular, they propose to skip the pre-fill stage of later layers by rewiring the model and rely instead on intermediate computation results from an earlier layer, leading to a reduced amount of computation. The authors also reduce the memory footprint of later layers by sharing a KV cache across multiple subsequent layers. Additional they use a distillation/fine-tuning process of the affected model part to reduce the difference in accuracy compared to the original model. They show in their evaluation computation improvements for throughput as well as latency.

**Strengths:**

The ideas for the various optimizations are presented reasonable clearly and they seem novel as well, especially their combination. The evaluation on a number of models and datasets/benchmarks supports their performance claims and a reasonable ablation study is provided as well.

**Weaknesses:**

For me, the biggest issue is that end-to-end results are missing, which makes it hard for me to put the presented inference results (throughput, latency) into context, which also makes me question how useful the presented numbers are.

* apart from SingleInputKV, all the other optimizations are not properly motivated regarding the reasoning why they should work (some form of microbenchmark)
* end to end results are missing, especially since some of their writing, if I am not mistaken, suggests that they target a part of the pipeline that only compromises 5% of the "runtime"
* line 314: "The accuracy drops by another 0.32, going from 2-way to 8-way" - it is actually 1.32 according to the table
  * similarly the number for 16-way is wrong as well
* The authors might consider removing 5.5 to have more space for presenting the other content/result in more detail.
* not clear why the used benchmarks are representative for the use cases mentioned as motivation in the introduction

detailed copy editing comments:
* related work:
  * "their optimized implementations in TensorRT (NVIDIA, 2019)" - all previously mentioned techniques were published after 2019
* Figure 2, right side:
  * The parameters used in the legend are not explained at all in the caption. It is possible to understand after reading the subsequent text.
  * The subfigure is never actually referenced in the text, except in the appendix, as "proof" for some statement later and in a footnote.
* Figure 4: Artifacts in the layering of the curves, sometimes a dot is at the top for one datapoint and and then further down for other datapoints. But maybe that was intentional?
* Table 3: There is a horizontal line missing after "(a) The effect of distillation".
* line 447: "which suggest that MLP layers player a more prominent role" - missing verb, probably it should be "play" instead of "player"
* Figure 5 is not readable
* minor issues:
  * typos:
    * line 119: "Tensor-Parallelism(Shoeybi et al., 2020)" - missing space
    * lines 130 to 138: additional brackets around the year for the citations
    * line 157/158: "(Holmes et al., 2024; Agrawal et al., 2024))" - additional bracket at the end
    * line 360: "toks/sec" - probably "token/sec"
    * line 859: "superme" - perhaps "supreme"?
    * line 923: "hyper-paramter" - missing e
    * line 923: "but did not invest deeper" - probably "investigated"
  * references:
    * Clark et al. 2018: cited differently than the other arXiv papers
    * Cobbe et al. 2021: misses place, where the paper was published
    * Dao et al. 2024:
      * year states 2024, but conference abbreviation suggests 2022
      * conference abbreviation is nowadays NeurIPS
    * Ding et al. 2021: misses place, where the paper was published
    * Elhage et al. 2021: url not clickable
    * GretelAI 2024: url not clickable
    * Hendrycks et al. 2021: cited differently than the other ICLR papers
    * Hinton et al. 2015: cited differently than the other arXiv papers
    * Kuzim et al. 2024:
      * year states 2024, but conference abbreviation suggests 2022
      * conference abbreviation is nowadays NeurIPS
    * Lewis et al. 2020: conference abbreviation is nowadays NeurIPS
    * Liu et al. 2024a:
      * cited differently than the other arXiv papers
      * cited twice (2024b)
    * Liu et al. 2024d:
      * year states 2024, but conference abbreviation suggests 2023
      * conference abbreviation is nowadays NeurIPS
    * Meng et al. 2024:
      * year states 2024, but conference abbreviation suggests 2022
      * conference abbreviation is nowadays NeurIPS
    * Pourreza and Rafiei 2024:
      * year states 2024, but conference abbreviation suggests 2023
      * conference abbreviation is nowadays NeurIPS
    * Sakaguchi et al. 2019: cited differently than the other arXiv papers
    * Wei et al. 2023: cited differently than the other arXiv papers

**Questions:**

* Section 3.4, Knowledge Recovery: The description suggests that the distillation is done for every of the later layers, but Figure 1 suggests that at least W_K and W_V are only trained for the initial layer of each AcrossKV block.
* Why are the results more or less consistently better for 4-way caching compared to 2-way caching for the 70B model? That seems kind of counterintuitive.
* footnote 5, page 7: What are the end-to-end results?
* Section 4.3: "a combined throughput of over 16K toks/sec over 4xH100 GPUs which corresponds to 560 TFLOPS/GPU"
  * So that is around 4k token per second for each GPU compared to 30k tokens/sec/GPU for 8B Llama model. But because the 70B model is much more complex, there are more floating point operations necessary?
  * Any notion why the pure compute performance increases despite a more "distributed" setting (multiple GPUs)?
* Any notion why the full model fine-tuning performs so much worse than the partial model fine-tuning?
* Section 5.3: "This may be due to the lack of math and coding examples in the two datasets we picked to train the model."
  * Why did you choose these datasets, if at least the coding use case is serving as a motivational example?
* Doesn't the discussion in Appendix B the whole point of the paper, i.e. trying to optimize a part than accounts for less than 5% of the total compute time?

---

> ### Author Response · Authors · 2024-11-19
> **Response to Reviewer Zy9U**
>
> Thank you for your detailed review of our paper and for your feedback. We will correct all of the copy editing suggestions in revising our paper.
>
> **On end-to-end results:** We would like to clarify the results presented in Section 4.3 (Figures 3 and 4) showcase the end-to-end performance of our work.  Specifically, these results highlight the performance of our final trained and transformed SwiftKV models, fully integrated into the vLLM system, which is the most widely used open-source serving system in industry. The measurements reflect the end-to-end throughput and latency improvements observed in our production system under simulated traffic.
>
> If the "end-to-end results" being requested refer to a different context or additional metrics beyond what is already presented, we kindly ask for clarification so we can address the concern more effectively.
>
> **On optimizing 5% of the total model:** To clarify, the 5% refers to the K and V projections of the later layers skipped by SingleInputKV that still need to be run. For example, with 50% SingleInputKV, we only need to run the KV projections of the last 50% of layers (skipping the Q, O, attention, and MLP operations), which amount to <5% of the later layers. We provide a full breakdown of the compute reductions in our top-level comment for all reviewers.
>
> **On Section 3.4, Knowledge Recovery:** When AcrossKV is used, only the Wk and Wv of the initial layer in each AcrossKV block is trained. We will clarify this in Sec 3.4.
>
> **Why 4-way caching is better than 2-way caching:** As stated in Appendix C.1, we did not search for the best hyper-parameters for different model sizes and/or different settings (e.g., the 4-way and 2-way caching) due to the limited resource and high cost to do so. As such, all the results we got are likely sub-optimal. Similar observation can be found in Table 1: the accuracy of 50% SingleKV + 4-way is higher than 2-way on GSM-8K.
>
> **On 16K tokens/s and 560 TFLOPs:** You are correct that the 70B model requires much more (approx 8-9x) floating point operations than the 8B model, and that is why the per-GPU tokens/s is much higher for the 8B model. The compute performance for the 70B model improves primarily due to the individual matrix multiplication operations being larger than for the 8B model, which is more suitable for the parallel GPU architecture. Even though the compute for 70B is distributed, it is still on the same node, which has very fast communication between GPUs.
>
> **Full model vs partial model fine-tuning:** We believe one reason that full fine-tuning performs worse is it can cause the model to overfit to the distillation dataset, which in SwiftKV’s case is very small compared to the scale of pretraining (or even instruction tuning). As stated in Appendix C.1, the total number of training tokens we have is 160M x 2 (epoch) = 320M. In Sec 5.2, we also mentioned prior works that hypothesize that the core knowledge of transformer models are stored in their MLP layers instead of Attention layers. Combined with the fact that SwiftKV only modifies the attention-related parts of the model, it makes sense that we should only train the Q, K, and V weights. However, if we are able to do the full scale pre-training and post-training (i.e., multi-iterations of SFT / DPO etc), the full model training might be better.
>
> **On dataset selection:** There is a very large search space of dataset mixtures that can lead to better accuracy, that would be infeasible to fully explore in the scope of our paper due to limited resources and high cost. Instead, we simply chose two very popular instruction datasets. Even though we could have optimized the dataset mixture to better fit the distribution of our evaluation tasks, we believe it can still be informative to readers to know the effect of dataset selection when it does not align well with the evaluation tasks. Therefore, we chose to use two basic datasets in our main evaluation, and provide insights into better dataset selection in our ablations (Sec 5.3). Nevertheless, it is both surprising and promising for SwiftKV that even a basic selection of data (with only 160M tokens) can lead to reasonable results.

---

> > ### Comment · Reviewer_Zy9U · 2024-11-25
> >
> > Thank you for your answers to my questions.

---

### Official Review · Reviewer_mqKq · 2024-11-03

**Soundness:** 2
**Presentation:** 2
**Contribution:** 2
**Rating:** 6
**Confidence:** 4

**Summary:**

This paper proposes two methods to reduce the cost of the prefill stage during LLM inference.
The first method, called SingleInputKV, reuses the output hidden state vector from the i-th layer in attention layers as the input vector to generate key and value (KV) vectors of the subsequent layers. In previous methods, the j-th layer used the output hidden state vector from the (j-1)-th layer as input.
The second method, called AcrossKV, enables the KV vectors generated by the i-th layer to be reused by the following layers. In previous methods, each layer generates its own KV vectors by multiplying the input vector with its weight.
These techniques reduce computational costs by reusing the input and KV vectors of earlier layers for later layers. They also decrease the number of KV vectors that need to be cached for the decode stage. The proposed methods build on prior work [1], which showed minimal differences in the values of input vectors across layers as the number of layers increases in transformers.
The authors implemented these techniques in Llama-3.1-8B and Llama-3.1-70B models,  showing that while the performance on the LLM benchmark remains largely unaffected, both time and memory usage in the prefill stage are reduced by almost two times.

[1] Songwei Liu, Chao Zeng, Lianqiang Li, Chenqian Yan, Lean Fu, Xing Mei, and Fangmin Chen. Foldgpt: Simple and effective large language model compression scheme, 2024c. URL https://arxiv.org/abs/2407.00928.

**Strengths:**

S1. The paper proposes two techniques derived from insights from prior research, demonstrating their efficacy in reducing computational and memory costs during LLM inference.

S2. The authors show that fine-tuning can alleviate the decline in benchmark scores, emphasizing the practicality of the proposed methods without notably sacrificing model performance.

**Weaknesses:**

W1. The experiments in the paper are somewhat limited.
- The authors evaluate the proposed techniques only on Llama-3.1 models. Testing a wider variety of models would strengthen the results. If the proposed methods could demonstrate their benefits across transformer models with different attention mechanisms (e.g., sparse attention, low-rank attention), scaling approaches (e.g., wide scaling, deep scaling, sparse scaling), and sizes (Llama-3.2-1B, Llama-3.2-3B, Llama-3.2-8B,  Llama-3.2.-11B, Llama-2-13b, and Llama-3.2-70B, Llama-3.2-90B, and Llama-3.1-405B), it would enhance the paper’s contribution.
- The authors need to demonstrate whether applying SwiftKV to larger models yields more significant results compared to small models. Incorporate models such as  Llama-2-13B and Llama-2-7B. If applying the proposed methods to Llama-2-13B yields better results than Llama-2-7B in terms of both cost and the benchmark scores, it would strengthen the contribution of the paper.
- There is no experiment to show the independent effect of AcrossKV without the presence of SingleInputKV, leaving the isolated impact of AcrossKV unexplored. The authors need to compare a baseline model to one with only AcrossKV applied.

W2. The justification for the claimed reduction in computational cost is insufficient. The paper needs to specify which operations are being skipped by SingleInputKV clearly by providing a detailed breakdown of the computational costs for each component of a Transformer model, comparing the baseline to SwiftKV. In Figure 1, SingleInputKV still appears to need to generate the output hidden state vector of every attention layer, which is a primary computational task in Transformer models. This is because the proposed method needs to generate the query vector for each attention layer in Figure 1, and the output hidden state vector from the (i-1)th layer is required to compute the query vector for the i-th layer.

**Questions:**

Please refer to W1 and W2.

---

> ### Author Response · Authors · 2024-11-19
> **Response to Reviewer mqKq**
>
> Thank you for your detailed review of our paper and for your helpful feedback.
>
> **On evaluating a wider range of models:** We provide the accuracy scores for two additional models, Mistral-Small-Instruct-2409 and Llama-3.1-405B-Instruct (FP8) in our top-level comment for all reviewers. For Mistral-Small, we observe similar impacts to model quality and throughput as Llama-8B. For Llama-3.1-405B-Instruct with W8A8 quantization, we additionally observe a smaller model quality gap, indicating that our method may be more robust in larger models. We are happy to include these results and their corresponding system performance results in revising our paper.
>
> **On computation cost reduction:** First, we would like to clarify that with SingleInputKV, only the decode tokens, not prefill tokens, need to calculate the attention outputs for all layers. For prefill tokens, if using 50% SingleInputKV, the last half of layers only need to calculate the key and value projections to populate the KV cache. This is done based on the output of the middle layer, not the immediate preceding layer, which allows the query projections, attention outputs, and MLPs to be skipped entirely for the last half of layers.
>
> As suggested, we provide a more detailed breakdown of the compute in our top-level comment for all reviewers, which we are happy to add to our revised paper.
>
> **On showing the independent effect of AcrossKV:** We emphasize that the key contribution of SwiftKV is prefill compute reduction (SingleInputKV) combined with KV cache memory reduction, and refer to our top-level comment for a detailed discussion. As demonstrated, even Merge-all-Layers, which can be considered an idealized version of AcrossKV, has limited throughput benefits if applied without SingleInputKV in compute-bound scenarios. Due to this fact, and the fact that KV cache memory compression has been explored thoroughly in prior works, we do not believe an in-depth evaluation of AcrossKV-only is crucial to our paper.

---

> > ### Comment · Reviewer_mqKq · 2024-11-23
> >
> > The authors’ response appears insufficient in addressing the concerns raised.
> >
> > First, the additional experiments presented in the response do not seem to confirm that SwiftKV performs well in scenarios involving attention mechanisms, scaling approaches, or across different model sizes.
> >
> > Second, I would like to see clear evidence in the revised paper that demonstrates using SwiftKV on a given model yields better results than using a smaller model, approximately half the size of the original.
> >
> > Third, please revise Figure 1 to clearly separate the prefill process from the distillation process. The current figure makes it difficult to understand how SingleInputKV and AcrossKV operate.
> >
> > Finally, I suggest including an evaluation and a deep analysis of performance when SingleInputKV is applied across layers where AcrossKV has been incorporated.

---

> > > ### Author Response · Authors · 2024-11-25
> > >
> > > We thank the reviewer for the clarifications on their concerns. We have performed additional evaluations, which we will add to our paper in our revisions.
> > >
> > > **On scenarios involving attention mechanisms, scaling approaches, or across different model sizes:** We provided results for Llama-405B-Instruct, a very large model, in our response to all reviewers. In addition, we would like to share accuracy evaluations for Llama-3.2-3B-Instruct and Deepseek-V2-Lite-Chat, which have diverse size, attention, and scaling characteristics:
> > >
> > > A summary of our additional findings:
> > > 1. **Llama-3.2-3B-Instruct (smaller model):** We find that SwiftKV maintains reasonable quality up to 40% of layers skipped via SingleInputKV, and observe a steeper drop-off at 50%, which is slightly earlier than the other larger models we tested. This finding is consistent with prior works’ observations that larger models may be more resilient to compression and pruning. On the other hand, this model is more resilient to AcrossKV (only 0.36 pt degradation from no AcrossKV to 4x AcrossKV).
> > > 2. **Deepseek-V2-Lite-Chat (mixture-of-experts and low-rank attention):** This model incorporates several unique architecture components, including fine-grained MoE with both shared and routed experts, and multi-head latent attention (MLA) that compresses the KV cache into a latent vector. Although MLA does not use the standard key- and value- projections, SwiftKV can still be adapted to it. We find that SingleInputKV performs well for this model at 0.6 pt degradation at 45% SingleInputKV, which is better than many other models in our evaluation. AcrossKV also obtains reasonable results at 2x AcrossKV, but suffers a larger degradation at 4x AcrossKV.
> > >
> > > **Llama-3.2-3B-Instruct**
> > > Benchmark | Baseline | 25% SingleInputKV | 40% SingleInputKV | 40% SingleInputKV + 2x AcrossKV | 40% SingleInputKV + 4x AcrossKV | 50% SingleInputKV
> > > |-|:-:|:-:|:-:|:-:|:-:|:-:|
> > > Arc-Challenge | 0.7517 | 0.7559 | 0.7534 | 0.7482 | 0.7559 | 0.7125
> > > Winogrande | 0.6859 | 0.6977 | 0.6898 | 0.6866 | 0.6921 | 0.6875
> > > Hellaswag | 0.7332 | 0.7234 | 0.7137 | 0.7141 | 0.7079 | 0.7077
> > > TruthfulQA | 0.5145 | 0.5280 | 0.5110 | 0.5067 | 0.5089 | 0.5129
> > > MMLU | 0.6201 | 0.6189 | 0.6180 | 0.6155 | 0.6135 | 0.5963
> > > MMLU-CoT | 0.6248 | 0.6239 | 0.6162 | 0.6103 | 0.6082 | 0.5994
> > > GSM8K-CoT | 0.7232 | 0.7111 | 0.6868 | 0.6777 | 0.6770 | 0.6702
> > > **Average** | **0.6647** | **0.6655** | **0.6555** | **0.6513** | **0.6519** | **0.6409**
> > >
> > > **Deepseek-V2-Lite-Chat**
> > > Benchmark | Baseline | 25% SingleInputKV | 45% SingleInputKV | 45% SingleInputKV + 2x AcrossKV | 45% SingleInputKV + 4x AcrossKV
> > > |-|:-:|:-:|:-:|:-:|:-:|
> > > Arc-Challenge | 0.6553 | 0.6544 | 0.6561 | 0.6552 | 0.6134
> > > Winogrande | 0.7466 | 0.7505 | 0.7395 | 0.7426 | 0.7521
> > > Hellaswag | 0.8156 | 0.8152 | 0.8082 | 0.8023 | 0.7980
> > > TruthfulQA | 0.5098 | 0.5053 | 0.5020 | 0.4985 | 0.4839
> > > MMLU | 0.5686 | 0.5691 | 0.5633 | 0.5559 | 0.5482
> > > MMLU-CoT | 0.5061 | 0.5092 | 0.5156 | 0.5051 | 0.3080
> > > GSM8K-CoT | 0.6869 | 0.6899 | 0.6611 | 0.6557 | 0.6489
> > > **Average** | **0.6412** | **0.6419** | **0.6351** | **0.6307** | **0.5932**
> > >
> > > *We use 45% SingleInputKV since it has 27 layers and we need a number of layers divisible by 4 for AcrossKV.
> > >
> > > **On SwiftKV vs using a smaller model of 50% the size:** We apologize for misunderstanding the reviewer’s original question. As suggested, we show accuracy results for Llama-2-7B-chat, Llama-2-13B-chat, and Llama-2-13B-chat with 50% SingleInputKV. As demonstrated below, Llama-2-13B-chat with 50% SingleInputKV attains a 5% higher average evaluation score than Llama-2-7B-chat.
> > >
> > > We also note similar gaps between smaller and larger models by other works. For example, [Minitron](https://arxiv.org/pdf/2407.14679v1) reports a 5% gap in MMLU and Hellaswag between their 4B and 8B models (Table 5). Meanwhile, the Minitron models are also more compute-intensive to train, requiring 94B tokens compared to SwiftKV's <1B tokens.
> > >
> > > Benchmark | 7B | 13B | 13B + 50% SingleInputKV
> > > |-|:-:|:-:|:-:|
> > > Arc-Challenge | 0.4352 | 0.5009 | 0.4676
> > > Winogrande | 0.7316 | 0.7569 | 0.7545
> > > Hellaswag | 0.7867 | 0.8250 | 0.8139
> > > TruthfulQA | 0.4558 | 0.4412 | 0.4385
> > > MMLU | 0.4384 | 0.5339 | 0.5346
> > > MMLU-CoT | 0.2310 | 0.3372 | 0.3180
> > > GSM8K-CoT | 0.2547 | 0.3783 | 0.3745
> > > Average | **0.4762** | **0.5390** | **0.5288**
> > >
> > > **On revising Figure 1:** We are happy to update our figure, and have uploaded a revised version to our supplementary materials for your review during the discussion phase. We will replace our main Figure 1 with this figure in revising our paper with the feedback from all reviewers.
> > >
> > > **On evaluation of performance when SingleInputKV is applied across layers where AcrossKV has been incorporated:** We kindly ask for further clarification on this suggestion. For all the evaluations in our paper and in our responses to reviewer feedback, SingleInputKV is indeed applied to the layers where AcrossKV is incorporated (AcrossKV merges the KV cache of layers generated by SingleInputKV).

---

### Official Review · Reviewer_DcFc · 2024-11-04

**Soundness:** 3
**Presentation:** 4
**Contribution:** 2
**Rating:** 5
**Confidence:** 4

**Summary:**

This paper studies the optimization of inference in Transformer-based LLMs. It presents SwiftKV, a solution to reducing the KV cache and inference time to long contexts up to 128K. The proposed method features three parts: SingleInputKV, AcrossKV, and knowledge recovery. Experiments show the effectiveness of the proposed method and the usefulness of its components, as well as how they work jointly with other optimization techniques.

**Strengths:**

S1. Inference optimization for in Transformer-based LLMs is an important topic which has been extensively studied in recent years.

S2. Several key components have been proposed in this paper, with their usefulness showcased in the evaluation.

S3. The proposed method is orthogonal to many existing optimizations and they can be used jointly to further optimize the performance.

**Weaknesses:**

W1. SingleInputKV borrows observations and ideas from previous works, as stated in the submission (such observation has also been utilized in the InfiniGen paper published at OSDI 2024).

W2. A core technique in the proposed method is cross-layer KV cache compression. The comparison/discussion with state-of-the-art KV cache compression/merging/cross-layer works is missing, e.g., PyramidKV and infini-attention. It is encouraged to discuss the difference and the novelty compared to existing KV cache compression techniques in the related work. Some surveys can be found here:
https://github.com/October2001/Awesome-KV-Cache-Compression

W3. Whereas the paper discusses long inputs, it lacks discussions with recent works on long contexts (see the above link), such as MInference, which optimize the prefilling of long contexts. Some of them exploit the sparsity to reduce KV cache and speed up inference, e.g., ALISA.

W4. The proposed method is not training free, yet only Llama-3.1 models are evaluated. It is unclear if the performance (and its optimal parameter settings) also translates to models. Extension to other open models, such as Mistral, would be beneficial to understanding the contributions of this work.

**Questions:**

Q1. In Table 1, there is a significant drop in performance for 8B model on the math dataset GSK-8K. I suppose this is the harder case, meaning that the proposed method may not work well for the case of small models on tasks demanding more logic and reasoning. An analysis on why this performance drop occurs would be interesting.

Q2. In Table 3, to show the impact of distillation, AcrossKV is disabled. However, to reduce KV cache, AcrossKV should be enabled for inference serving, right?

Q3. In Table 4, the performance of "our fine-tuned model" is significantly inferior to the base model. The result seems to be negative to the usefulness of your techniques. I don't quite understand the logic here.

**Details Of Ethics Concerns:**

This paper studies LLM core technology. I don't find anything that needs ethics review in this paper.

---

> ### Author Response · Authors · 2024-11-19
> **Response to Reviewer DcFc**
>
> Thank you for your detailed review of our paper and for your feedback.
>
> **Response to W1 and W2:** Indeed, several independent works including ours made observations regarding the similarity of activations across transformer layers. However, we are not aware of any work that leverages this insight to reduce the amount of pre-fill computation to speed up inference. InfiniGen reduces the KV cache memory needed during the decoding phase to reduce data transfer between CPU and GPU to speed up heterogeneous inference with CPU offloading, however it does not reduce the prefill computation required.
>
> As we shared in our comment for all reviewers, the key novel contribution of SwiftKV is not KV cache compression alone but rather prefill compute reduction (SingleInput KV) combined with KV cache compression. Approaches such as InfiniGen and PyramidKV that only reduce KV cache memory have very little impact on the throughput in production environments where there is ample fast GPU memory. In contrast, the compute reduction from SingleInputKV significantly improves throughput.
>
> We appreciate the pointers to the additional related works on KV cache reduction and will add a more thorough discussion to our related works in revising our paper.
>
> **Response to W3:** Unlike MInference or ALISA that focuses on reducing only the quadratic attention computation and KV cache memory, SingleInputKV reduces the total computation in the prefill stage of the model, not just the attention computation. SingleInputKV is agnostic to the sparsity pattern used in attention and as such SingleInputKV is complimentary and can be combined with MInference, ALISA, or any other sparse attention mechanism to achieve a compound reduction in total inference compute. For instance, if MInference or ALISA can reduce overall inference compute by 2x, then combining it with a 2x reduction from SwiftKV will result in a compounded reduction of 4x.
>
> We also would like to point out that we showed SwiftKV can be combined with KV cache quantization in Sec 5.4. However, combining our method with all existing methods and showing the improvement is beyond the scope of a research work.
>
> **Response to W4:** We provide the accuracy scores for two additional models, Mistral-Small-Instruct-2409 and Llama-3.1-405B-Instruct (FP8) in our top-level comment for all reviewers. For Mistral-Small, we observe similar impacts to model quality and throughput as Llama-8B. For Llama-3.1-405B-Instruct with W8A8 quantization, we additionally observe a smaller model quality gap, indicating that our method may be more robust in larger models. We are happy to include these results and their corresponding system performance results in revising our paper.
>
> **Answer to Q1:** Indeed, GSM8K appears to be the task we evaluated that has the largest gap, which may be due to the small size of the model or higher demand for reasoning. In Sec 5.3, we explored another likely explanation, which is simply that our distillation can benefit from more math and coding related data. We found that using an additional 83K math and coding examples (only 7% of our existing dataset) improved the GSM8K score by 0.53 points. Although it is not in the scope of our paper to identify the optimal distillation dataset (also as stated in Sec 5.3), we believe future work in this direction can help to bridge the gap for GSM8K. Possible directions included better post-training selection, advanced training approaches (like DPO), and pre-training enhancement which are also stated in our Limitation and Future Direction section.
>
> Also, as the question itself stated: there is a noticeable gap of LLaMa-3.1-8B but the gap reduces to ~1.5 points for LLaMa-3.1-70B with the same training data and hyper-parameters. This indicates a larger model may be more resilient to compression, and this is also illustrated by the 405B cases in the above table, where the gap further reduces to 0.9 points.
>
> **Answer to Q2:** Given the main goal of SwiftKV is to reduce the prefill compute rather than KV cache memory, we felt that including AcrossKV was not critical to illustrate the effect of our distillation procedure on compute reduction. However, we are happy to take the suggestion and include a row for AcrossKV in Table 3 in revising our paper.
>
> **Answer to Q3:** “Our fine-tuned model” here is not a SwiftKV model, but a model fine-tuned on (not distilled from) Llama-3.1-8B. The purpose of the “Our fine-tuned model” row serves only to illustrate the difference in data quality between what was used to distill SwiftKV and what Meta (presumably) used to create Llama-3.1-8B-Instruct.
>
> Our message is that, since there is a large quality gap between the model we fine-tuned ourselves versus Meta’s instruct model, there is likely a large quality gap in the datasets used to fine-tune them. This further illustrates (as in our answer to Q1) that improving the distillation dataset quality is likely a promising path to improving SwiftKV in future work.

---

> > ### Comment · Reviewer_DcFc · 2024-11-20
> > **Re: Author Response**
> >
> > Thank the authors for the response, especially for the effort to conduct additional experiments. The response addressed most of my concerns. I would like to raise my score. For the revised version, I believe a discussion with recent studies may help understand the positioning of this work, as there have been many papers targeting this problem recently.

---

> > > ### Author Response · Authors · 2024-11-20
> > >
> > > Thank you for reading our response and providing your feedback. We will add more substantial discussions about related works as suggested. Based on your scores, we were curious if this is the only remaining reservation, or if there are others. If you could share them, we would be happy to try to address then during the discussion period. Please let us know.

---

> > > > ### Comment · Reviewer_DcFc · 2024-11-21
> > > > **Re: Author Response**
> > > >
> > > > Yes, considering that ICLR is a conference that encourages novelty, a 5-rating was given because SingleInputKV shares the same idea with previous works on this topic (though they don't target prefilling), compromising the novelty of this work. Another major drawback is that distillation is required. In contrast, MInference, which also optimizes prefilling, is training-free, making it easier to deploy in real-world applications.

---

> > > > > ### Author Response · Authors · 2024-11-22
> > > > >
> > > > > We greatly appreciate the clarification of your remaining concerns about our paper and for allowing us a chance to address your concerns!
> > > > >
> > > > > **On the novelty of SwiftKV:** Like our related works, SwiftKV certainly builds upon the shoulders of its predecessors. Particularly, the fact that hidden states between deeper layers can be highly similar has been observed by multiple works before InfiniGen as well (https://arxiv.org/pdf/2403.19135 and https://arxiv.org/pdf/2310.17157 Fig 5b), which we also leverage in SwiftKV.
> > > > >
> > > > > We do not believe that sharing this widely accepted observation about LLMs detracts from the novelty of our work. SwiftKV’s novelty primarily lies in its goal to focus on prefill computation reduction and its novel architecture re-wiring that enables it, which is substantially different from any prior work. Additionally, SwiftKV is the only work to our knowledge that not only reduces the attention computation, but also the Q, O, and MLP of prefill tokens.
> > > > >
> > > > > **On the distillation requirement:** While SwiftKV does require distillation, we would like to highlight that the scale is substantially smaller than prior works that also leverage distillation. As mentioned in our top-level comment, LayerSkip (https://arxiv.org/pdf/2404.16710) and Minitron (https://developer.nvidia.com/blog/how-to-prune-and-distill-llama-3-1-8b-to-an-nvidia-llama-3-1-minitron-4b-model/) are distilled on 10-100B tokens, while SwiftKV utilizes only 320M tokens. This makes SwiftKV easily trainable in a few hours using a single node for the 8B model, or even on commodity hardware.
> > > > >
> > > > > While recent works show that pruning and layer skipping can be done using a similar or smaller scale distillation as SwiftKV, their effectiveness is typically much lower. For example, LLM-Pruner (https://arxiv.org/pdf/2305.11627) suffers a 5-10% quality gap even at 20% pruning ratio, while Shortened Llama (https://arxiv.org/pdf/2402.02834) suffers a 7-11% quality gap at 35-37% pruning ratio. Meanwhile, SwiftKV suffers only a 2% quality gap at 50% of layers skipped for prefill tokens.
> > > > >
> > > > > Thus, SwiftKV lies at a unique junction of having a lightweight distillation requirement, high compute reduction, and small model quality gap, which is different from all preceding work. This is due to our careful decisions to focus on prefill computation, design of the SingleInputKV model transformation, and precise selection of model parameters to train during distillation. We consider the combination of these decisions as another significant novel contribution of our work.
> > > > >
> > > > > **On comparison with MInference:** Although MInference is training-free, we disagree that it automatically means it is easier to deploy in production. MInference (and many other sparse attention works) require custom attention kernels to run efficiently on GPUs. Integration between these custom kernels and existing inference systems can be highly non-trivial, and is not addressed by MInference. For example, PagedAttention (https://arxiv.org/pdf/2309.06180) is the core system technique in vLLM that allows it to achieve low memory fragmentation and high throughput. However, MInference’s integration with vLLM simply overrides the entire attention module with its own (https://github.com/microsoft/MInference/blob/main/minference/patch.py#L1082), making it impossible to use the current implementation of MInference and PagedAttention at the same time.
> > > > >
> > > > > On the other hand, SwiftKV requires no custom kernels to run with existing inference systems, and is largely agnostic to the mechanics of the attention operation. In fact, SwiftKV should be fully compatible with MInference as well. Integration into existing inference systems such as vLLM is a key contribution of our work (Sec 3.5), which we used for all of our system performance experiments and will open-source to the community so it can be easily run by others for real-world applications.
> > > > >
> > > > > In addition, optimizations that only target the attention operation are only impactful for very long sequence lengths, as evidenced by MInference targeting 100K - 1M sequence lengths and illustrated by our compute breakdown in our comment for all reviewers. SwiftKV works for all sequence lengths because it equally reduces the compute of the Q, O, and MLP operations for all prefill tokens.
> > > > >
> > > > > Thank you for the very helpful discussion of our work, and we will make sure this positioning is clear in revising our paper.

---

> > > > > > ### Comment · Reviewer_DcFc · 2024-11-22
> > > > > > **Re: Author Response**
> > > > > >
> > > > > > **Novelty**: SingleInputKV is based on an observation which has appeared in previous works . As such, the observation itself is not new. Though it is applied to the optimization of prefilling in this paper, this still belongs to the category of fast inference of long contexts, in which the observation has been utilized before. In other words, the employment of this observation is immediate: it has been used for decoding (e.g., InfiniGen), can we use it for prefilling?
> > > > > >
> > > > > > **Deployment**: I agree that the reported number of tokens for distillation is small. Nonethelss, the users' effort is that they need to prepare the dataset for the distillation, and this step is non-trivial. Your experimental results also suggest that improving the distillation dataset quality is worthy of exploration.
> > > > > >
> > > > > > **MInference**: I don't agree with the claim that "(it is) impossible to use the current implementation of MInference and PagedAttention at the same time". In fact, MInference has [adapted itself to PagedAttention](https://github.com/microsoft/MInference/blob/main/minference/modules/minference_forward.py#L881). You may also find that vllm>=0.4.1 is supported in its [document](https://github.com/microsoft/MInference). It even supports FlashAttention-2, showcasing that its custom attention kernels can jointly work with prevalent inference optimization tools.

---

> > > > > > > ### Author Response · Authors · 2024-11-22
> > > > > > >
> > > > > > > We thank the reviewer for the quick responses as it makes for an enjoyable and engaging discussion!
> > > > > > >
> > > > > > > **On novelty of SwiftKV:** We believe we have presented our case and evidence. We agree with the reviewer that SwiftKV builds upon prior observations about the hidden state similarity between deeper layers, which we clearly acknowledged in Sec 3.2. We do not believe this detracts from the novelty of SwiftKV, which lies in its model architecture transformation and carefully designed distillation procedure that enables high prefill compute reduction, low accuracy degradation, and small training scale. However, we understand that evaluations of novelty can be subjective and we respect the reviewer’s opinions on this matter.
> > > > > > >
> > > > > > > **On deployment vs MInference:** We thank the reviewer for the correction and pointer to the code. It appears the integration is incomplete as the prefix-enabled attention code path still uses the standard PagedAttention without MInference. This code path is required for chunked-prefill (i.e. [Split-Fuse](https://arxiv.org/pdf/2401.08671), [Sarathi](https://arxiv.org/pdf/2308.16369)) which is now enabled by default in vLLM for long-context models (https://github.com/vllm-project/vllm/pull/6666). Moreover, although FlashAttention-2 is supported, [FlashAttention-3](https://github.com/Dao-AILab/flash-attention) has been available for some time.
> > > > > > >
> > > > > > > We understand that these new developments occurred after the publication of MInference, and thus are not reasonable to be supported. Although conceptually possible, some amount of effort still needs to be made to integrate them with sparse attention (SwiftKV is also conceptually compatible with sparse attention). On the other hand, SwiftKV is largely agnostic to kernel-level and system-level compatibility issues as it simply re-wires the same set of model operations.
> > > > > > >
> > > > > > > To the reviewer’s original point that, “in contrast, MInference, which also optimizes prefilling, is training-free, making it easier to deploy in real-world applications”, we believe our discussion nevertheless reveals a more nuanced trade-off. For users who are experienced in kernel- or system-level programming, maintaining ongoing compatibility with sparse attention techniques such as MInference may indeed be a reasonable (or even preferred) choice. For others who are more experienced in modeling, SwiftKV offers a more system-agnostic approach.

---

### Author Response · Authors · 2024-11-19
**Comment for all reviewers**

**We thank the reviewers for their thoughtful and detailed feedback.**

We emphasize that the key novel contribution of SwiftKV is not KV cache memory compression alone but rather *prefill compute reduction (SingleInputKV)* combined with KV cache compression. While prior works address KV cache compression, we are not aware of those that target compute reduction for prefill-only via layer skipping. Memory compression without compute reduction (prior works on KV compression) can be helpful in scenarios with limited GPU memory, but it has limited impact in production environments with A100/H100 GPUs where GPU memory is sufficient and inference is compute-bound.

To illustrate, we benchmark the throughput of an “ideal” KV compression scheme, where every layer’s KV cache is merged into a single layer (Merge-all-Layers). We retain the computation for all KV operations (i.e., W_kX and W_vX) but eliminate the memory for all layers > 1, leading to a single layer of KV cache. Merge-all-Layers represents a “best case compression scenario” with:
- Extreme compression ratio beyond any published technique, e.g. 32x and 80x for Llama-3.1 8B and 70B, respectively.
- Zero overhead, while most techniques (e.g., quantization, low-rank decomposition) add extra computations or data conversions.

Model | Input length | Baseline (tokens/s) | Merge-all-Layers (tokens/s) | 50% SingleInputKV (tokens/s) |
| -- | -- | :-: | :-: | :-: |
Llama-3.1-8B (1x H100) | 4K | 18.3K | 19.8K | 25.2K
| | 16K | 16.2K | 17.0K | 23.5K
Llama-3.1-70B (4x H100) | 4K | 9.19K | 10.1K | 13.2K
| | 16K | 8.21K | 9.12K | 12.8K

The ideal merge-all-layers marginally improves baselines, while SwiftKV with 50% SingleInputKV alone obtains much stronger improvements. As there is already sufficient GPU memory to support a large enough batch to saturate the GPUs, memory compression does not yield significant improvements. Thus, compute reduction, rather than memory compression, is crucial for improving performance in compute-bound scenarios.

**Prefill compute breakdown:** To further illustrate SwiftKV’s prefill compute reduction, we provide an example breakdown of Llama-3.1-70B model operations (theoretical GFlops per prefill token for 128K context).

Model | Vocab | Q | K | V | O | MLP | ATTN | Total per Prefill Token | Prefill Compute
| -- | -- | :-: | :-: | :-: | :-: | :-: | :-: | :-: | :-: |
Baseline | 4.3 | 11 | 1.3 | 1.3 | 11 | 113 | 160 | 301 | 100%
25% SingleInputKV | 4.3 | 8.1 | 1.3 | 1.3 | 8.1 | 85 | 120 | 228 | 75.5%
50% SingleInputKV | 4.3 | 5.4 | 1.3 | 1.3 | 5.4 | 56 | 80 | 154 | 51.1%
50% SingleInputKV + 4x AcrossKV | 4.3 | 5.4 | 0.84 | 0.84 | 5.4 | 56 | 80 | 153 | 50.8%

First, 25% and 50% SingleInputKV reduces the total prefill compute by 24.5% and 48.9%, respectively. SingleInputKV allows most of the compute-heavy operations (Q, O, MLP, ATTN) to be skipped entirely for the later layers. Second, SingleInputKV skips the attention operation as well to retain its relative performance improvements for long and short inputs alike.

**More model evaluations:** Llama-3.1-405B-Instruct (FP8) and Mistral-Small-Instruct-2409 (22B)

Mistral-Small achieves similar model quality results as Llama-8B in our paper (~1% gap for 50% SingleInputKV). Llama-405B is a much larger model we quantized with W8A8. The gap for Llama-405B is smaller (at ~0.7%). This indicates that SwiftKV may be more robust for larger models. We also achieved consistent throughput improvements for both models.

**Llama-3.1-405B-Instruct (FP8)**

Model Quality

Benchmark | Baseline | 50% SingleInputKV |
| -- | -- | -- |
Arc-Challenge | 94.7 | 94.0
Winogrande | 87.0 | 86.3
Hellaswag | 88.3 | 88.1
TruthfulQA | 64.7 | 64.2
MMLU | 87.5 | 85.7
MMLU-CoT | 88.1 | 87.5
GSM8K-CoT | 96.1 | 95.2
**Average** | **86.6** | **85.9**

Throughput

Input Length | Baseline (tokens/s) | 50% SingleInputKV (tokens/s)
| - | :-: | :-: |
2K | 4417 | 5958
8K | 4229 | 5958
32K | 3364 | 4792
128K | 1744 | 2492

**Mistral-Small-Instruct-2409 (22B)**

Model Quality

Benchmark | Baseline | 50% SingleInputKV  | 50% SingleInputKV + 4x AcrossKV
| - | :-: | :-: | :-: |
Arc-Challenge | 84.1 | 83.5 | 82.9
Winogrande | 84.7 | 84.0 | 83.8
Hellaswag | 87.3 | 86.3 | 86.2
TruthfulQA | 56.8 | 55.6 | 56.0
MMLU | 73.3 | 72.9 | 72.3
MMLU-CoT | 74.9 | 74.0 | 73.0
GSM8K-CoT | 86.5 | 84.3 | 82.5
**Average** | **78.2** | **77.2** | **76.7**

Throughput

Input Length | Baseline (tokens/s) | 50% SingleInputKV (tokens/s)
| - | :-: | :-: |
2K | 6962 | 8479
8K | 6362 | 8123
32K | 4276 | 5690

**Lightweight training:** Lastly, SwiftKV training is very light weight. All SwiftKV models are trained using about 160M x 2 epochs = 320M tokens, far less than other training based methods like early exit [[1]] (26B-52B tokens) and/or pruning [[2]] (100B tokens) but achieves better accuracy with similar computation reduction. We will highlight this in revising our paper as well.

[1]: https://arxiv.org/pdf/2404.16710
[2]: https://arxiv.org/pdf/2407.14679v1

---

### Author Response · Authors · 2024-11-28
**Thank you to Reviewers**

We would like to thank all of our reviewers for their insightful comments and dedication in reviewing our work and engaging with us during the discussion period. We believe that the feedback given by each reviewer helped to greatly improve the quality and scope of our work.

Although the time for new experiments, discussions, and revisions is limited, we would like to summarize our major additional results and our revision progress in the discussion period thus far.

1. Discussion about the primary contribution of SwiftKV being prefill compute reduction combined with KV cache reduction, rather than KV cache reduction alone. Experimental evidence that shows compute reduction is crucial for compute-bound scenarios. We have added this to Sec 5.1 of our revised draft.
2. Additional model quality evaluations on Llama-3.2-3B-Instruct, Llama-3.1-405B-Instruct (FP8), Mistral-Small-2409 (22B), and Deepseek-V2-Lite-Chat, which cover a more diverse set of model sizes (3B - 405B), attention mechanisms (MLA), and scaling (MoE) **[Reviewer mqKq,DcFc]**. We have added this to Sec 4.2 of our revised draft.
3. Ablation using Llama-2-7B and Llama-2-13B that showed Llama-2-13B with SwiftKV achieves higher accuracy than Llama-2-7B at roughly the same 50% prefill compute reduction **[Reviewer mqKq]**. We plan to add this to our paper after the review period.
4. Implementation of SwiftKV on SGLang, a second and different inference system than vLLM, along with experiments that show SwiftKV’s performance improvements translates from vLLM to SGLang **[Reviewer 2qP6]**. We plan to add this to our paper after the review period.
5. Ablation study that shows AcrossKV’s performance impact in low-memory settings **[Reviewer 2qP6]**. We plan to add this to our paper after the review period.
6. Throughput benchmarks using more realistic workload traces derived from ShareGPT **[Reviewer 2qP6]**. We plan to add this to our paper after the review period.
7. Clearer breakdown of the prefill compute reductions resulting from SwiftKV **[Reviewer mqKq,Zy9U]**. We have added this as Table 1 in our revised draft.
8. Clearer discussion on related works: memory compression techniques, and sparse attention systems with reference to ALISA and MInference **[Reviewer DcFc]**. We have added this to the related works of our revised draft.
9. Clearer discussion about the distillation scale of SwiftKV compared to other works that employ distillation **[Reviewer DcFc]**. We plan to add this to our paper after the review period.
10. Clearer figure that illustrates the prefill process and compute reduction **[Reviewer mqKq]**. We have uploaded a revised Figure 1.

We also thank the reviewers for pointing out important discussions and copy-editing mistakes in our draft. We plan to add and address all of these in revising our paper after the review period.

We hope the additional experiments and discussions are helpful for the reviewers in evaluating our work, and we are happy to address any additional feedback or questions in the remaining time during the discussion period.

Sincerely,

Submission13422 Authors

---

### Meta-Review · Area_Chair_JUuq · 2024-12-22

**Metareview:**

While SwiftKV presents a solid engineering approach to optimizing LLM inference through KV cache optimization and model transformation (which I totally appreciate the effort especially the opensourced release), it (as the reviewers suggested) falls slightly below the bar for ICLR. Though the work demonstrates meaningful performance improvements, several limitations prevent acceptance: the core techniques largely build upon existing methods, the experimental evaluation is limited, and there's insufficient comparison with state-of-the-art KV cache compression techniques.

**Additional Comments On Reviewer Discussion:**

I have read the messages in the discussion period and my opinion has been summarized as in the metareview above. I considered these points in my recommendation.

---

### Decision · Program_Chairs · 2025-01-22

Reject